# Mapping cortical mesoscopic networks of single spiking cortical or sub-cortical neurons

Dongsheng Xiao[1,2,3†], Matthieu P Vanni[1,3†], Catalin C Mitelut[1,3†], Allen W Chan[1,3], Jeffrey M LeDue[1,3], Yicheng Xie[1,3], Andrew CN Chen[2], Nicholas V Swindale[4], Timothy H Murphy[1,3*]

[1]Department of Psychiatry, Kinsmen Laboratory of Neurological Research, Vancouver, Canada; [2]Beijing Institute for Brain Disorders, Capital Medical University, Beijing, China; [3]Djavad Mowafaghian Centre for Brain Health, University of British Columbia, Vancouver, Canada; [4]Department of Ophthalmology and Visual Sciences, University of British Columbia, Vancouver, Canada

**Abstract** Understanding the basis of brain function requires knowledge of cortical operations over wide-spatial scales, but also within the context of single neurons. In vivo, wide-field GCaMP imaging and sub-cortical/cortical cellular electrophysiology were used in mice to investigate relationships between spontaneous single neuron spiking and mesoscopic cortical activity. We make use of a rich set of cortical activity motifs that are present in spontaneous activity in anesthetized and awake animals. A mesoscale spike-triggered averaging procedure allowed the identification of motifs that are preferentially linked to individual spiking neurons by employing genetically targeted indicators of neuronal activity. Thalamic neurons predicted and reported specific cycles of wide-scale cortical inhibition/excitation. In contrast, spike-triggered maps derived from single cortical neurons yielded spatio-temporal maps expected for regional cortical consensus function. This approach can define network relationships between any point source of neuronal spiking and mesoscale cortical maps.

*For correspondence: thmurphy@mail.ubc.ca

[†]These authors contributed equally to this work

**Competing interests:** The authors declare that no competing interests exist.

## Introduction

Neural activity ranges from the microscale of synapses to macroscale brain-wide networks. Meso-scale networks occupy an intermediate space and are well studied in cortex forming the basis of sensory and motor maps (*Bohland et al., 2009*). These networks are largely defined by co-activation of neurons and have been evaluated with a variety of statistical approaches that capitalize on detecting synchrony. The study of large-scale networks (meso-to macroscale) has been mostly restricted to functional magnetic resonance imaging (fMRI), or magnetoencephalography that can capture whole-brain activity patterns (*de Pasquale et al., 2010*; *Kahn et al., 2011*; *Logothetis et al., 2012*), but lack high spatial and temporal resolution and sensitivity. To overcome these limitations, alternative approaches including mesoscopic intrinsic signal, voltage, glutamate, or calcium sensitive indicator imaging have been employed (*Kleinfeld et al., 1994*; *Kenet et al., 2003*; *Ferezou et al., 2007*; *Chemla and Chavane, 2010*; *Chen et al., 2013b*; *Mohajerani et al., 2013*; *Stroh et al., 2013*; *Vanni and Murphy, 2014*; *Carandini et al., 2015*; *Chan et al., 2015*; *Madisen et al., 2015*; *Wekselblatt et al., 2016*; *Xie et al., 2016*). New preparations using large-scale craniotomies (*Kim et al., 2016b*) and large format imaging systems (*Tsai et al., 2015*; *Sofroniew et al., 2016*) provide the ability to link mesoscale activity patterns to individual neurons. However, these measures are restricted to superficial layers of cortex and cannot assess linkages with sub-cortical structures.

Developments in fiberoptic technology allow local optical functional assessment of brain activity in sub-cortical structures (*Hamel et al., 2015*; *Kim et al., 2016a*), but cannot simultaneously resolve cortex over large fields of view. Although the evolution of imaging has revealed new aspects of cortical processing in identified neurons (*Harvey et al., 2012*; *Chen et al., 2013a*, *Chen et al., 2013b*; *Fu et al., 2014*; *Guo et al., 2014*), the electrically recorded action potential is still a signal of prominence given its exquisite timing and ability to reflect the output of neuronal networks (*Buzsáki, 2004*).

We combine extracellular recordings of single units in the cortex, thalamus, and other sub-cortical sites with mesoscopic functional imaging in transgenic mice expressing the calcium indicator GCaMP (*Zariwala et al., 2012*; *Vanni and Murphy, 2014*; *Silasi et al., 2016*). While slower than protein-based or small molecule voltage sensors, GCaMP imaging offers a high signal-to-noise ratio and is associated with supra-threshold activity which allows a more direct comparison with spike activity. This work extends pioneering studies investigating the relationship between single neuron spiking and local neuronal population activity assessed by voltage-sensitive dye imaging. Spike-triggered averaging (STA) was used to identify the local activity profile related to the spiking activity of a single neuron within this population (*Arieli et al., 1995*), and it was further demonstrated that this activity profile could reveal the instantaneous spatial pattern of ongoing population activity related to a neuron's optimal stimulus in visual cortex of anesthetized cat (*Tsodyks et al., 1999*). This current study extends these approaches and also exploits the main advantage of mesoscopic imaging allowing the simultaneous measurement of brain activity in multiple regions across most of cortex simultaneously and not only that of the local population of activity surrounding the recording site. This multiscale strategy has allowed us to define temporal relationships between the activity of single neurons at the microscopic scale and mesoscale cortical maps (*Zingg et al., 2014*; *Madisen et al., 2015*). Furthermore, we employ multisite, long shank, silicon probe recordings of single neuron activity that facilitates the assessment of long-distance activity relationships between remote subcortical single neuron activity and mesoscale cortical population activity. Spontaneous activity in awake and anesthetized mice was exploited as a source of diverse cortical network activity motifs (*Mohajerani et al., 2010*, *2013*; *Chan et al., 2015*). Application of STA to cortical spontaneous activity linked single neurons to mesoscale networks. Single thalamic neuron spikes were found to functionally link to multiple primary sensorimotor maps, in contrast spiking cortical neurons were largely associated with consensus cortical maps. Thalamic neurons were found to both predict and report (firing before and after) specific cycles of wide-scale cortical inhibition/excitation, while cortical neuron firing was usually associated with excitation. These results are consistent with an active computational role of thalamus in sensory-motor processing (*Theyel et al., 2010*; *Hooks et al., 2013*; *Petrus et al., 2014*; *Sheroziya and Timofeev, 2014*; *McCormick et al., 2015*), as opposed to merely serving a relay function and is consistent with a diverse role of the thalamus in feedforward sensory processing. Thalamocortical transmission can dynamically and differentially recruit local cortical excitation and inhibition based on thalamic neuron firing patterns and where thalamocortical feedforward inhibition is a critical feature (*Galarreta and Hestrin, 1998*; *Swadlow and Gusev, 2001*; *Gabernet et al., 2005*; *Cruikshank et al., 2007*; *Hu and Agmon, 2016*). We expect that this spike-triggered cortical mapping technique, exploiting mesoscopic calcium imaging, can be extended to any brain location where electrodes can be placed to identify functionally linked cortical mesoscale networks.

## Results

### Linkage of individual spiking neurons to specific mesoscopic cortical maps

We exploit the wide field of view of mesoscale cortical imaging using GCaMP transgenic mice (*Madisen et al., 2015*) in combination with cellular electrophysiology recordings to derive cortical networks that reflect activity at targeted point sources of neuronal spiking throughout the brain. Cortical and sub-cortical neuron spiking activities were recorded electrically while simultaneously imaging cortical mesoscopic activity across a 9 × 9 mm bilateral window that encompassed multiple areas of the mouse dorsal cortex including somatosensory, motor, visual, retrosplenial, parietal association and cingulate areas (*Figure 1A,B*). Spectral decomposition of the mesoscopic spontaneous

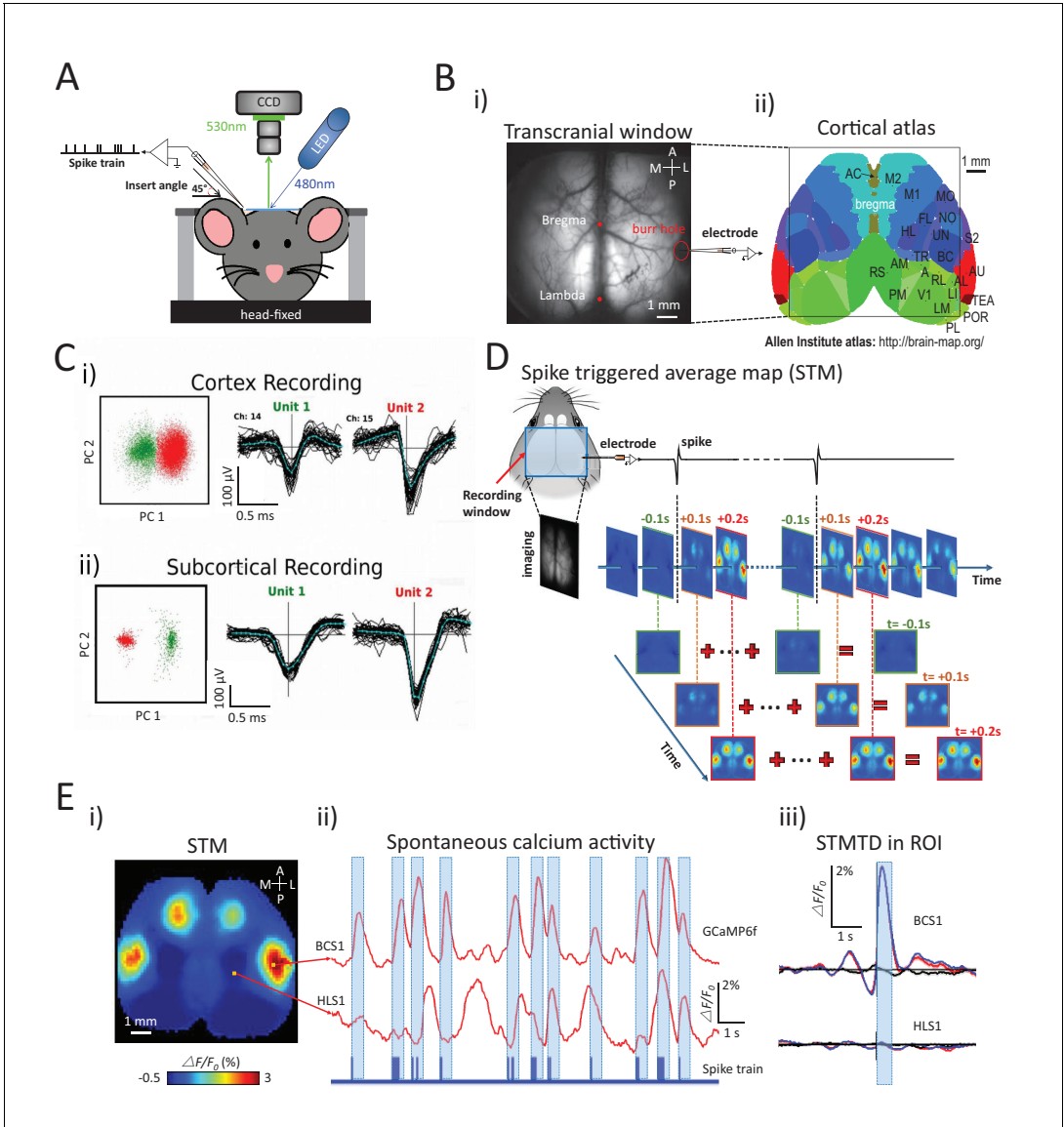

**Figure 1.** Experimental setup and multichannel electrode recordings and spike classification. (A) Set-up for simultaneous wide-field calcium imaging and single unit recording using a glass pipette or laminar silicon probe. (Bi) Top view of wide-field transcranial window and (ii) cortical atlas adapted from the Allen Institute Brain Atlas. (C) Example of (i) cortical and (ii) subcortical pairs or spike recordings from separate channels showing the isolation in the two principal components axes. (D) The generation of a spike-triggered average map (STM) for unit located in barrel cortex. (Ei) STM generated from single neuron with 1158 spikes recorded in right barrel cortex. (ii) *Red traces:* Spontaneous calcium activity recorded from two different cortical areas (BCS1 and HLS1). *Blue trace:* spontaneous spiking activity recorded simultaneously from right BCS1. (iii) STMTD generated from average of calcium activity time-locked with each spike (red) and random spike (see Materials and methods, black, blue: subtraction of spike and random spike-evoked responses) in region-of-interest (ROI). These examples results were from mice under anesthesia. Source files for the generation of spike-triggered average map (STM) can be found at http://goo.gl/nHF29I. The folder 'Matlab code and source data' (see the sublink of https://github.com/catubc/sta_maps) contains calcium images ('tif' file), spike train and Matlab code used for the generation of STM shown in *Figure 1E.* The 'tif' file cannot not be viewed with a standard picture viewer, but must be viewed with a program, such as 'ImageJ'. Spike times were exported as 'txt' file. Matlab code (named 'STA_eLife.m') was used for reading images and spike time files and generating STM.

The following figure supplements are available for figure 1:

**Figure supplement 1.** Spectral distribution of the spontaneous activity.

**Figure supplement 2.** Deconvolution analysis of GCaMP data.

activity using GCaMP6 revealed the presence of information below 10 Hz that was distinct from non-specific green light reflectance (*Figure 1—figure supplement 1*). Given the slow $Ca^{2+}$ binding and unbinding kinetics of GCaMP6, we expect imaging dynamics will be prolonged compared to actual spike records. In some cases, we employed deconvolution (*Pnevmatikakis et al., 2016*) to improve the time course of raw calcium signals (*Figure 1—figure supplement 2*). While deconvolution improved the temporal dynamics of the decay of the calcium signal, spike-triggered analysis was only marginally affected and it was not used throughout. Spiking signals were initially recorded in multiple brain areas using glass electrodes (n = 8 mice) to minimize obstruction of cortical imaging and reduce potential for damage from electrode placement. Subsequently, laminar probes (16 channel with 0.1 mm contact spacing) permitted the resolution of more spiking neurons simultaneously, and facilitated the recordings in multiple sub-cortical regions (n = 16 mice). Given the invasive nature and the long duration of recordings, initial data were obtained from urethane (n = 4) or isoflurane (n = 12) anesthetized adult mice, but were later optimized to include awake recordings (n = 12, see Materials and methods). The spike-triggered average maps (STMs) obtained under both these conditions were qualitatively similar, and this observation was consistent with previous work using VSD imaging (*Mohajerani et al., 2013*). To perform these assessments, we identified single neuron spikes from extracellular recordings using spike sorting methods based on clustering of principal components distributions of spike signals on sets of adjacent channels (*Swindale and Spacek, 2014*) (*Figure 1C*).

To investigate how single neuron spiking activity at a cortical/sub-cortical site was related to regional cortical activity, we calculated STMs from simultaneously acquired wide-field calcium imaging (*Figure 1D*; see also Materials and methods). For each individual spike, we considered cortical image frames from 3s before to 3s after the spike normalized as $\Delta F/F_0$ by subtracting and dividing the average calcium activity during the 3s preceding the spike. The static STM was then defined as the peak response (in units of $\Delta F/F_0$) calculated for each pixel within a time window of ±1 s. This peak method is a better reflection of correlated activity than the average over the ±1 s interval which would smooth out highly activated – but short duration activity increases. This method revealed that the activity recorded from a single right barrel cortex neuron yielded an STM showing strong and specific GCaMP signal in the barrel and motor cortices of both hemispheres (*Figure 1Ei*). STMs were thus calculated by averaging calcium activity during spiking activity (*Figure 1Eii*) and revealed the high spatial specificity of the mapping when compared with reference region (hind limb) or random spike averaging (see Materials and methods) (*Figure 1Eiii*).

We verified the calcium specificity of STMs (reflecting underlying neuronal activity) by imaging Thy-1 GFP-M mice (n = 6 mice) that lacked calcium-dependent neuronal fluorescent signals and failed to produce functional maps using the same procedures (*Figure 2A,B*). To investigate the sensitivity of the technique, the minimum number of spikes needed to make maps in GCaMP mice was measured by quantifying the similarities of pairs of STMs generated from a subset of increasing numbers of spikes. Stable STM maps were generally observed using 256 spikes (*Figure 2C–E*). A high stability of STMs was also confirmed by comparing the maps generated by splitting a unit's spikes into two halves, or into odd and even groups which yield similar STMs.

## Thalamic neurons show more diverse STMs than cortical neurons

By combining wide-field calcium imaging and single unit recording, the mesoscopic network associated with any neuron of the brain can be mapped. After establishing the method, we then focused on thalamic and cortical recordings of spiking neurons and confirmed their anatomical location by labeling probes with Texas red-dextran or DiI to visualize tracks (*Figure 3A*, four experiments sub-cortical track and nucleus identification, see also *Figure 2A,B*). We observed that spiking cortical neurons were linked to consensus local and long-range cortical networks as defined by the stability of STMs between recordings (*Figure 3B*). We define consensus cortical networks as those that can be identified by assessment of correlated activity (seed pixel analysis) and reflect major mono-synaptic intra-cortical axonal projections (*Mohajerani et al., 2013*). Spiking barrel cortex neurons were consistently linked to regional GCaMP signal changes in the barrel and motor cortex, as well as showing signals in homotopic areas of both hemispheres, consistent with the previously observed pattern of long-distance connections (*Ferezou et al., 2007*; *Mohajerani et al., 2013*; *Guo et al., 2014*; *Vanni and Murphy, 2014*; *Chan et al., 2015*). To compare multiple neuron STMs, we displayed the 'contour' of each STM as the full-width-half-max value (*Figure 3B,C*-contours).

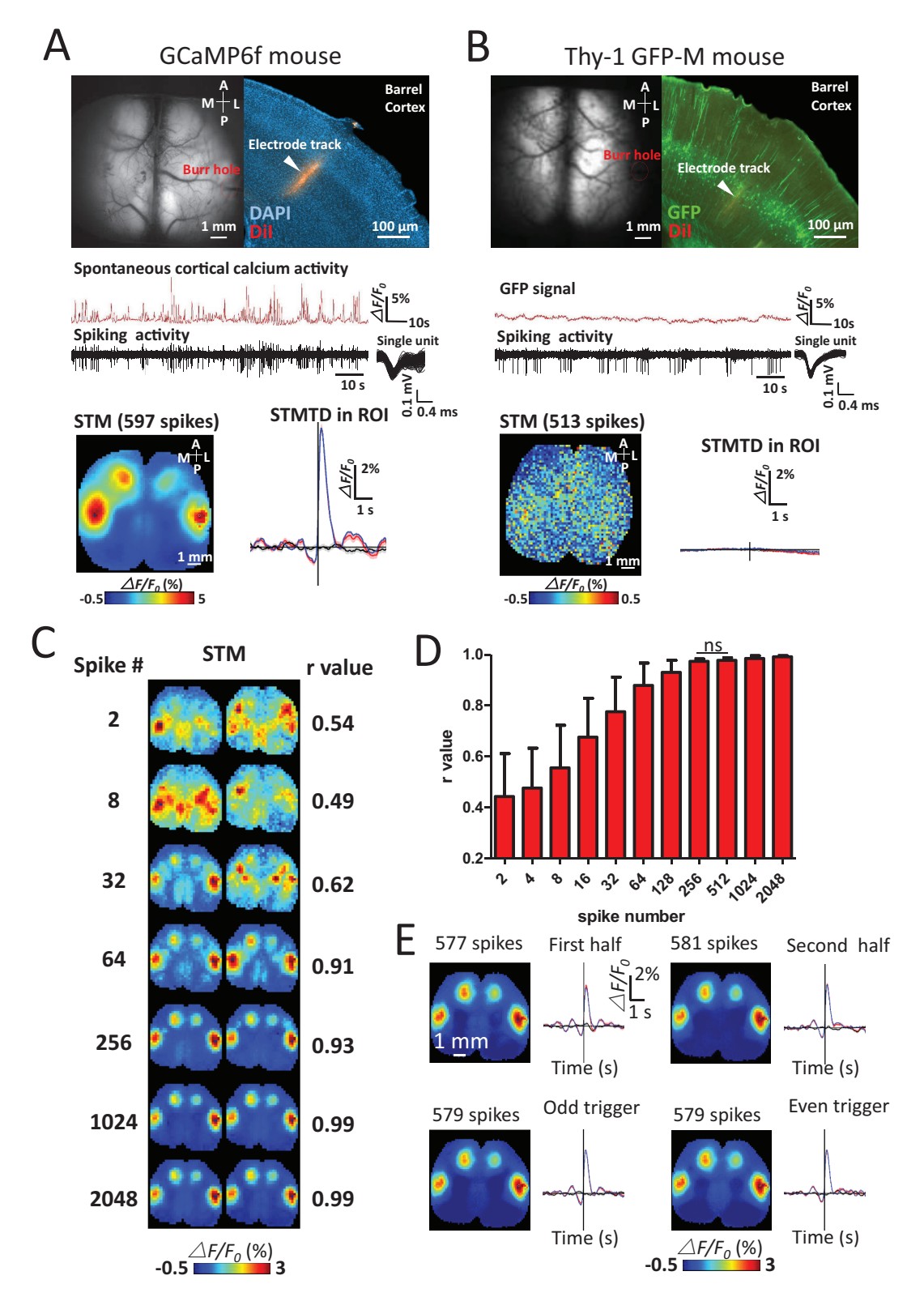

**Figure 2.** Sensitivity and specificity of STMs. (**A**) Simultaneous calcium and spiking activity recording in GCaMP6f mouse and STM yielded from single unit recorded in barrel cortex. (**B**) Simultaneous GFP fluorescence and spiking activity recording in Thy-1 GFP-M mouse and STM yielded from single unit recorded in barrel cortex resulted in no clear regional map. (**C**) STMs generated from a subset of spikes (2–2048, on the left) randomly chosen in one experiment. Correlation coefficients (r-value on the right) between STMs were used to evaluate the consistency of mapping. In this example, STMs

*Figure 2 continued on next page*

*Figure 2 continued*

generated by more than 64 spikes generated a correlation >0.9 and revealed high similarity between the pairs of SPMs made using the same number of spikes. (**D**) Distribution of correlation values between pairs of STM for an increasing number of spikes. No significant change in r-value distribution was observed for 512 spikes in comparison to 256 spikes (Mann Whitney test, p=0.126, U = 948.5, 256 spikes group n = 58, r-value = 0.97 ± 0.01, mean ± SD; 512 spikes group n = 40, r-value = 0.98 ± 0.01). (**E**) STMs and profile of responses computed using spikes divided into halves or even-odd sets. These examples were performed under anesthesia.

We have also assessed other means of generating event triggered maps including Multi-Unit-Activity (MUA) and local field potential (LFP) frequency bands. Cortical and thalamic STMs computed from single unit activity or MUA strongly resembled one another and did not vary greatly according to laminar depth. LFP-triggered STMs for delta band activity were similar to STMs, while those associated with higher frequency bands showed more unique patterns that will be investigated in future work (*Figure 3—figure supplement 1* and *Figure 3—figure supplement 2*). Delta band activity is where most mesoscale power functional imaging indicator power is located (*Chan et al., 2015*). In contrast, in the case of GCaMP6 higher frequency components are closer to hemodynamic and other noise sources making analysis more challenging.

STMs for single thalamic neurons indicated not only a functional link between GCaMP maps and their consensus cortical projection areas (*Hunnicutt et al., 2014*; *Oh et al., 2014*; *Zingg et al., 2014*) but also showed more variability and complexity in behavior than for cortical units. Thalamic neurons were associated with both unilateral and bilateral hemispheric signals within multiple primary sensorimotor and higher order brain areas (*Figure 3C*). To quantify the difference of variability of STM between cortical and thalamic units, we compared the percentage overlap of static STMs (*Figure 3D*; red areas) for neighboring pairs of neurons (100 µm apart). We found cortical neurons in the same functional region exhibited substantial similarity, while subcortical neurons had more diversity even within the same sub-nucleus in thalamus (i.e. VPL, VPM, see *Table 1* for nomenclature and compare 3 example in *Figure 3C*). The use of multichannel probes allowed us to obtain spiking profiles across cortical layers. However, there were only subtle topographic changes: the contours of static STMs were largely similar between and within superficial and deep layers, respectively (see the contour map in *Figure 3B*). To assess diversity in single neuron spiking derived cortical maps (STMs), we compared STMs derived from neighboring electrode contacts and found that sub-cortical derived maps were more varied (*Figure 3Diii*)

## Sub-cortical neurons are linked to cortical maps not predicted from consensus networks

To determine quantitatively how cortical/sub-cortical STMs were related to intra-cortical networks, static STMs were compared using cross-correlation with a cortex-wide library of seed pixel correlation maps (SPM) (*Mohajerani et al., 2013*; *Vanni and Murphy, 2014*; *Chan et al., 2015*) generated iteratively for all locations from the same recording of spontaneous activity (*Figure 4A,B*). To create SPMs, the cross-correlation coefficient r values between the temporal profiles of one selected pixel and all the others within the field of view were calculated (*Mohajerani et al., 2013*; *Vanni and Murphy, 2014*; *Chan et al., 2015*) (see Materials and methods). To evaluate the similarity between the static STM and SPMs, we calculated the correlation coefficient between pixels of both types of maps for all possible SPMs in the library. We then selected the SPM that resulted in best match: highest correlation between a given STM and the library of SPMs. The library of SPMs is expected to reflect cortical consensus activity motifs (areas undergoing temporally-correlated activity) and can be largely attributed to their underlying intra-cortical axonal projections (*Mohajerani et al., 2013*). Single cortical neuron-derived STMs were largely predicted by the pattern of cortical connectivity using SPM correlation mapping (*Figure 4A*) as correlations were relatively high between these SPMs and STMs (*Figure 4C*). In contrast, thalamic STMs were more complex and corresponded to more unique distributions of cortical patterns and had significantly lower correlations with the cortical consensus SPM library (*Figure 4B,C*). For example, VPM, VPL and CP neurons can functionally link to multiple cortical areas that are not predicted by SPM (SPM made by putting the seeds in either BCS1, HLS1 or RS areas). In other words, it is possible that the cortical STMs derived from single spiking sub-cortical neurons are the super-position of 2 or more cortical networks defined by SPMs. To support this

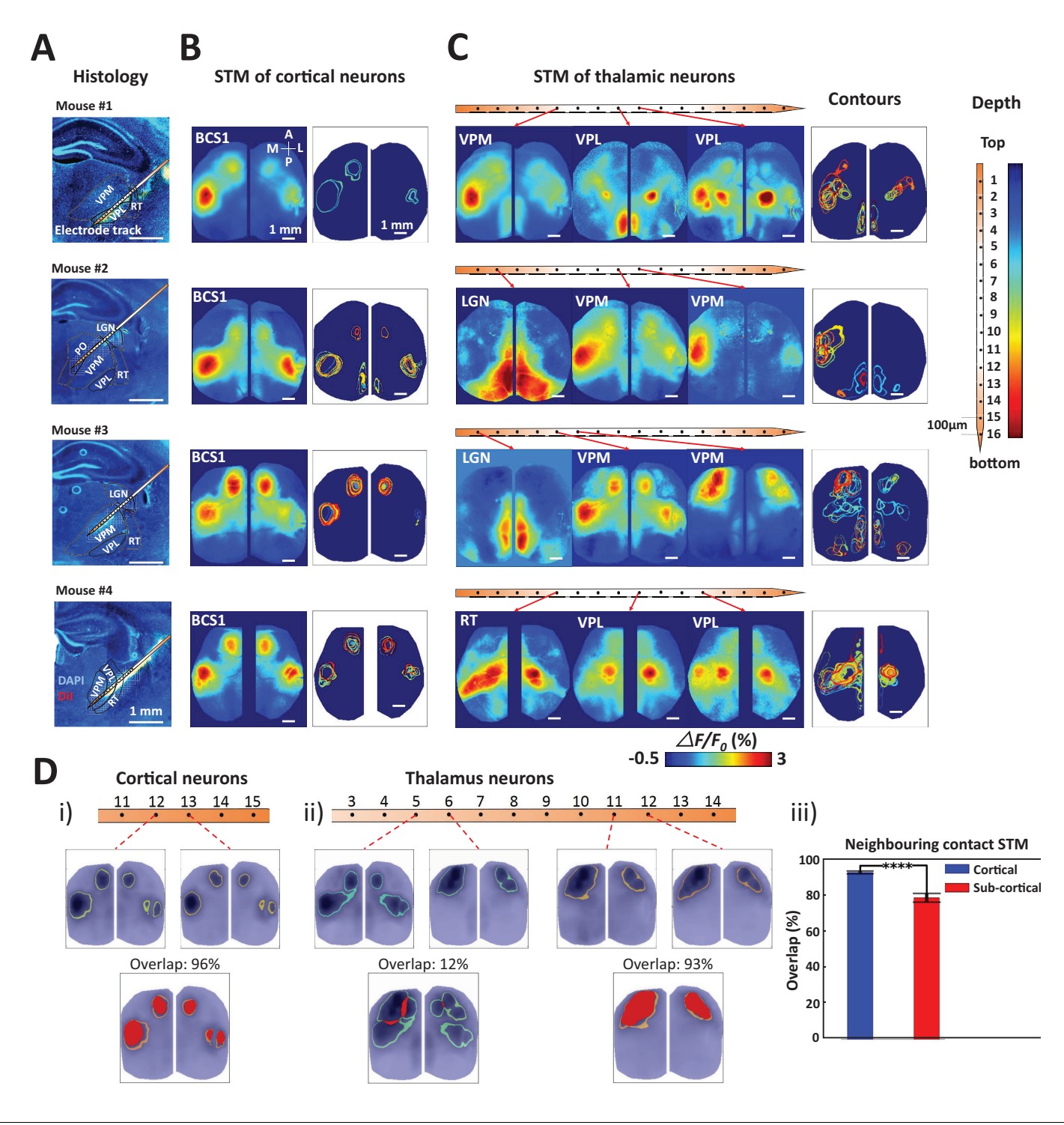

**Figure 3.** Topographic properties of cortical and thalamic STM. (**A**) Electrode track for each recording (Blue channel: DAPI, yellow: DiI). (**B**) STM and overlay contours of neurons recorded in barrel cortex in the same animal. Each color represents one STM contour. (**C**) STM and overlay contours of neurons recorded in thalamus in the electrode track presented in panels **A** and **B**. Color bar in the right side indicated the depth of each recording site. (**D**) Diversity of overlap of STMs between neurons on neighboring laminar electrode channels. (**i**) Example of overlapping STMs (red area) between two cortical neurons recorded on adjacent channels. (**ii**) Example of overlapping STMs for neighboring pairs of neurons recorded subcortically showing differences across depth. (**iii**) Average neighboring cortical neuron map overlap (blue: 93%) and neighboring sub-cortical neuron overlap (78%) show significant differences (Mann Whitney test, p<0.0001, U = 617408.0, mean percentage overlap of cortical STM pairs = 92.77% ± 0.23%, mean ± SEM,
*Figure 3 continued on next page*

*Figure 3 continued*

n = 966; mean percentage overlap of sub-cortical STM pairs = 78.11% ± 0.61%, mean ± SEM, n = 1936). These results are from awake mice (except Mouse #1).

The following figure supplements are available for figure 3:

**Figure supplement 1.** STMs were computed at different depths of the electrode using single cell spikes, Multi-Unit-Activity (MUA) and local field potential (LFP) amplitude.

**Figure supplement 2.** Thalamic STMs: Spike vs LFP.

**Table 1.** Abbreviation used to define different cortical/sub-cortical areas.

| | |
|---|---|
| S1 | Primary somatosensory area |
| S2 | Supplemental or Secondary Somatosensory area |
| FL | Forelimb region of the Primary Somatosensory area (FLS1) |
| HL | Hindlimb region of the Primary Somatosensory area (HLS1) |
| BC | Barrel region of the Primary Somatosensory area (BCS1) |
| M1 | Primary motor area |
| M2 | Secondary motor area |
| MO | Mouth region of the Primary Somatosensory area |
| NO | Nose region of the Primary Somatosensory area |
| TR | Trunk region of the Primary Somatosensory area |
| UN | (Unassigned) region of the Primary Somatosensory area (S1) |
| AC | Anterior Cingulate area (ACC) |
| A | Anterior or Posterior Partial Association areas: PTLp or PTA |
| V1 | Primary visual cortex |
| AL | AnteroLateral regions of the extrastriate visual areas |
| AM | AnteroMedial regions of the extrastriate visual areas |
| LM | LateralMedial regions of the extrastriate visual areas |
| PL | PosteroLateral regions of the extrastriate visual areas |
| LI | LateralIntermediate regions of the extrastriate visual areas |
| PM | PosteroMedial regions of the extrastriate visual areas |
| POR | Postrhinal regions of the extrastriate visual areas |
| RL | RostroLateral regions of the extrastriate visual areas |
| AU | Primary Auditory area |
| TEA | Temporal Association area |
| RS | Retrosplenial area |
| PTA | Parietal Association area |
| VPM | Ventral posteromedial nucleus of the thalamus |
| VPL | Ventral posterolateral nucleus of the thalamus |
| PO | Posterior complex of the thalamus |
| RT | Reticular nucleus of the thalamus |
| LGN | Lateral geniculate nucleus |
| CP | Caudoputamen |
| HPF | Hippocampal formation |

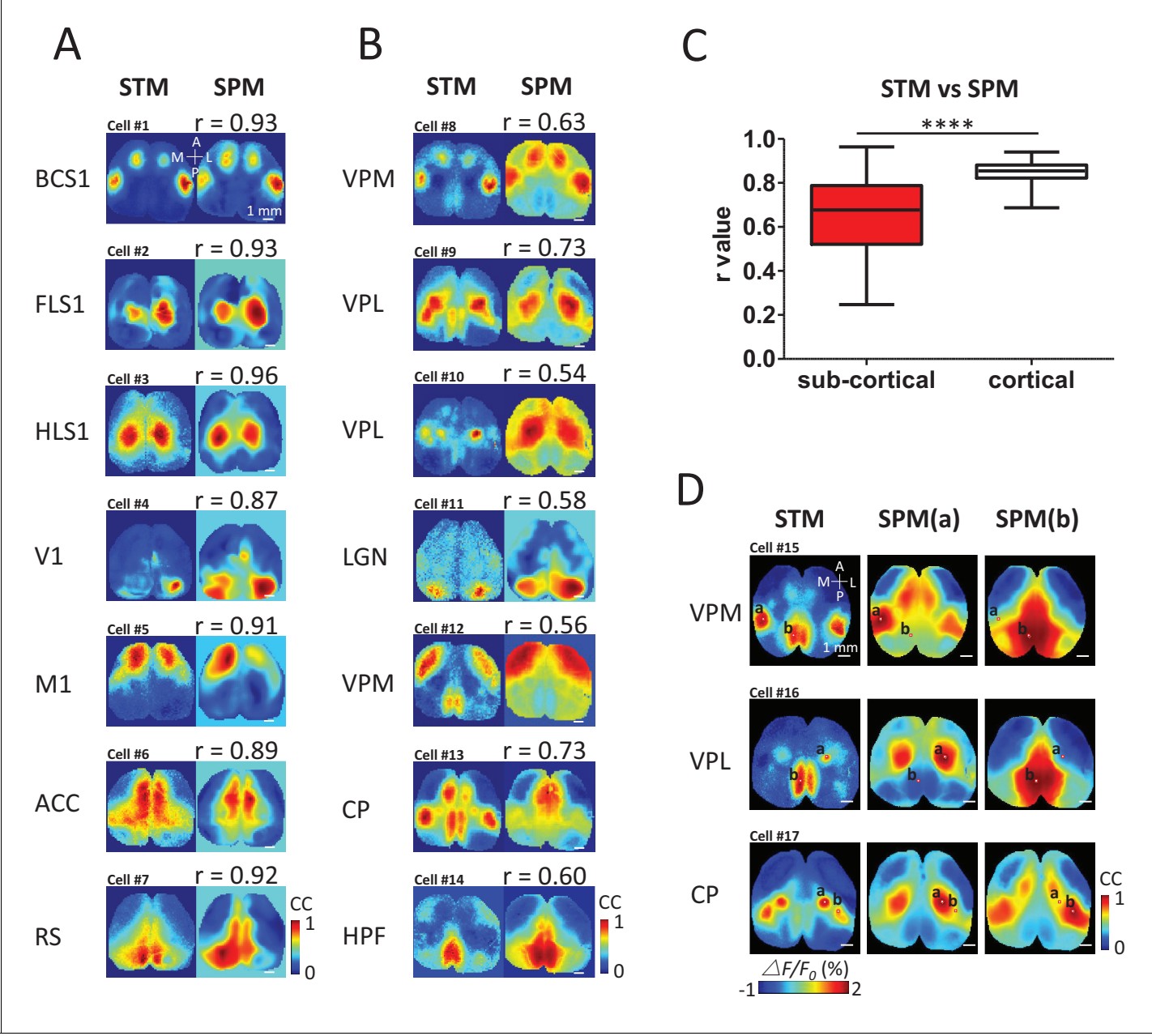

**Figure 4.** STM compared with seed pixel correlation maps (SPM). (**A**) Cortical STM (left) and the best fitting SPM (right) according to correlation coefficient (cc) values for different electrode placements (text to the left of panel). Similarity was calculated by measuring the r-value Pearson coefficient between each pair of map pixels (in title). Group data from 12 GCaMP6f mice are reported in panel **C**. (**B**) Sub-cortical STM (left) and the most similar SPM (right). Cells #2 and #4 were from GCaMP6s mice. Cells #3, 5–7, 11 were from GCaMP3 mice. Other cells were from GCaMP6f mice. These examples were performed under anesthesia. (**C**) Distribution of r-values (Mann Whitney test, p<0.0001, U = 5227, sub-cortical group n = 246 r-value=0.64 ± 0.18, mean SD; cortical group n = 168 r-value=0.85 ± 0.04, mean ± SD). (**D**) Examples of sub-cortical STMs compared with pairs of SPMs for seed indicated by 'a' and 'b'.

theory, we show examples comparing subcortical STMs to pairs of SPMs made from two seed locations that can lead to these potentially more complex maps (*Figure 4D*). For example, the VPM STM could be constructed as a combination of a BCS1 and RS SPM (seed point a and b, respectively, *Figure 4D*).

To better understand the underlying structural circuit basis of distinct thalamic STMs, we examined the Allen Mouse Brain Connectivity Atlas (*Oh et al., 2014*) as in our previous work (*Figure 5*) (*Mohajerani et al., 2013*). We processed the three-dimensional structural data and matched the composed anatomical 2D maps with our two-dimensional static STMs. As expected, we found that STMs of spiking cortical neurons correspond with underlying structural axonal projections (see BCS1 example in *Figure 5*, [*Mohajerani et al., 2013*]). However, sub-cortical STMs cannot be predicted by direct monosynaptic projections from sub-cortical to cortical areas. For example, the HPF has no strong direct structural link to RS area. Furthermore, CP was not directly linked to BCS1/HLS1 areas (*Hunnicutt et al., 2014*; *Oh et al., 2014*), indicating that sub-cortical STMs reflect apparently polysynaptic links to cortex.

## Cortical and sub-cortical neuron firing is tuned to cortical network dynamics spanning millisecond to multi-second time scales

By analyzing the cortical GCaMP signal, time courses corresponding to single neuron firing (firing= time 0) we observed dynamic cortical activity states (*Figures 6* and *7*; *Videos 1–4*). We determined STM Temporal Dynamics (STMTD) by identifying a region-of-interest (ROI) and tracking the time course of the maximally activated/depressed cortical pixel of the region from 3s before to 3s after spiking. As all our extracellular recordings were in right barrel cortex and predominantly right sensory thalamus, we identified the left barrel cortex as a co-activated area of interest and tracked dynamics within this ROI for spiking neurons. As observed for static STMs, the cortical calcium dynamics associated with spiking cortical neurons were relatively homogeneous with an initial peak in activity within ~100 ms – 200 ms following spiking and a return to baseline (*Figure 6A*). However, some cortical cells (~20%) participated in multi-second depression dynamics (see distribution of profiles in *Figures 7B,C and* and *8A*). In contrast, STMTDs generated by thalamic cells were more varied and were dominated by depression dynamics (~80% of cells) lasting up to 3 s (*Figures 6B* and *7B,C* and *Figure 8B*).

Our analysis indicates that averaging GCaMP cortical motifs from all spikes of a single cell produces converging STMs and STMTDs. However, thalamocortical synapses are known to be prone to synaptic depression and burst pattern firing may yield altered cortical responses (*Castro-Alamancos, 1997*; *Gil et al., 1997*). It is conceivable that averaging all spike activity may mask STMs from sub-groups of spikes that contribute different motifs during varying ongoing cortical dynamics. One way to divide the spikes into groups is to partition them into bursting versus tonic modes. We found that both STMs and STMTDs are similar across spiking modes to the all-spike average STMs in both cortical and thalamic cells examined (*Figure 6—figure supplements 1*, *2* and *3*; see Materials and methods). We also tried grouping spikes according to motif similarity but did not find natural groupings or clusters. Manually partitioning the motifs into groups of four or more, and removing the spontaneous motif average, revealed that the sub-group STMs were the same as the overall average (*Figure 6—figure supplements 4* and *5*; see Materials and methods). These tests confirm that the averaging method reveals how single cells participate in largely stereotyped networks despite the variability of ongoing cortical activity.

While our analyses have employed spike-triggering averaging, spike-triggered co-variance (*Aljadeff et al., 2016*) or variance maps are another way to view the association between GCaMP activity and spiking as opposed to the mean. Spike-triggered variance maps revealed similar brain regions as defined using spike-triggered average (STA) mapping when examined as a difference to baseline (*Figure 6—figure supplement 6*). It is conceivably that during particular behavioral states or tasks that variance mapping may reveal markedly different maps than STA.

We next investigated whether the multi-second corticocortical and thalamocortical dynamics revealed by this novel functional approach can be categorized into distinct groups (*Figure 7*, *Figure 7—source data 1*). Identification of the location of maximally activated cortical regions provided a means of comparing cortical and thalamic neurons (*Figure 7A*). Using Principal Component Analysis (PCA) we plotted the distribution of 428 STMTDs from both cortical and sub-cortical cells (*Figure 7B*). The time courses were then clustered in three patterns of STMTDs (using k-means algorithm) and averaged. Based on their temporal relationship to spikes, we termed them: **pattern #1**: spike-triggered-excitation; **pattern #2** spike-triggered inhibition; **pattern #3** inhibition-triggered spiking followed by inhibition (*Figure 7Ci*). The distribution of these three patterns were then compared between cortical and thalamic cells (*Figure 7Cii*).

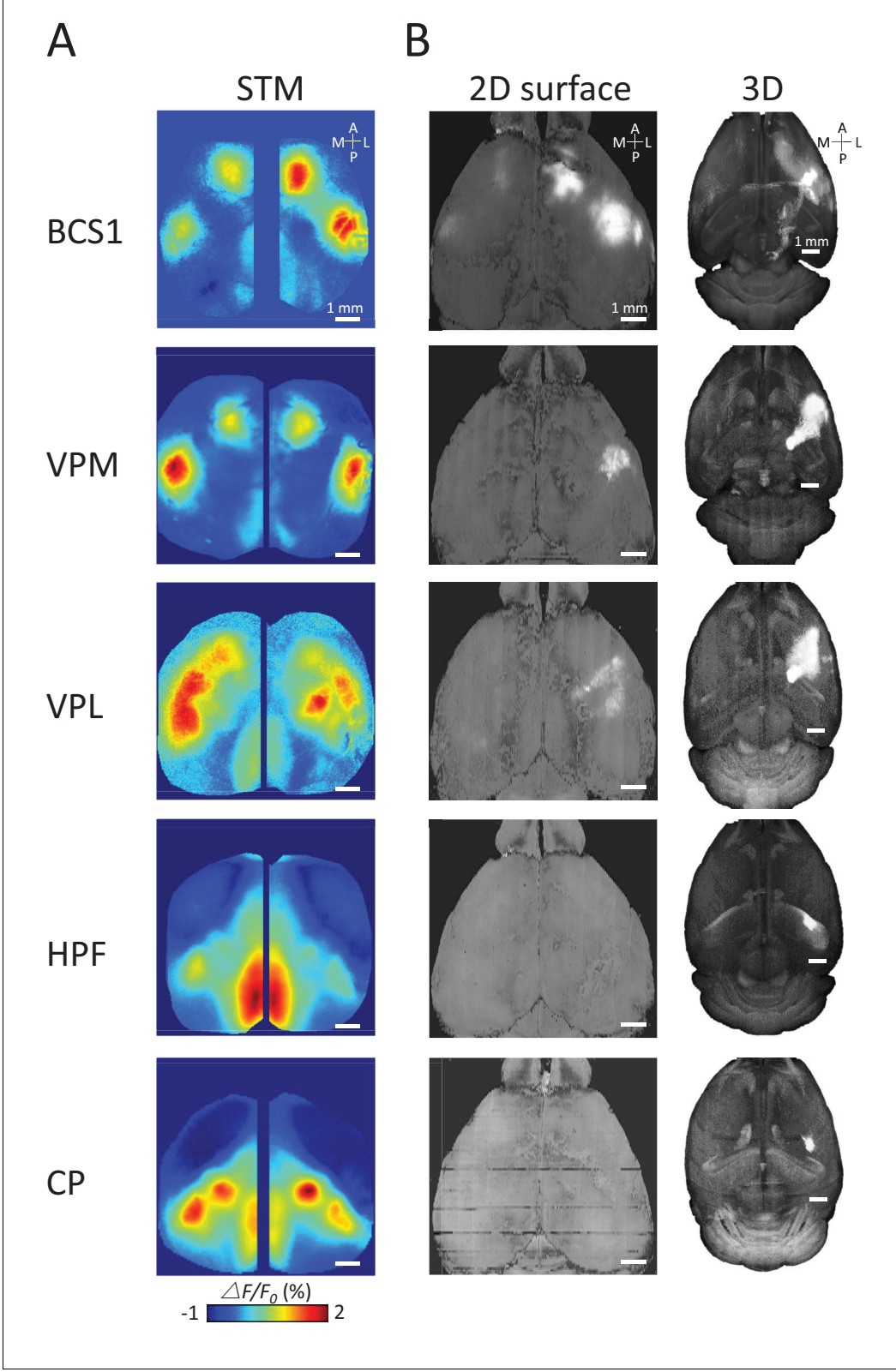

**Figure 5.** Examples of STMs and projection maps. (**A**) Example of STMs from neurons recorded in BCS1, VPM, VPL, HPF and CP. (**B**) Example of projection maps (2D surface and 3D) reconstructed from Allen Brain Atlas with injection sites (*Oh et al., 2014*) in the same region as our recording. For example, for a spiking neuron recorded in BCS1 we show anterograde labeling of GFP emanating from an injection site in BCS1 that extends to motor

*Figure 5 continued on next page*

*Figure 5 continued*

cortex and is present across cortex in the 2D surface plot of cortex. This projection pattern for BCS1 matches the STM map quite well as in previous work (*Mohajerani et al., 2013*). In contrast, for sub-cortical injections of GFP tracer such as in HPF there was less overlap between STMs and projection maps perhaps indicating polysynaptic pathways. Website: 2015 Allen Institute for Brain Science. Allen Mouse Brain Connectivity Atlas [Internet]. Available from: http://connectivity.brain-map.org.

This classification showed that ~80% cortical neurons were associated with a pure cortical excitation profile (pattern #1) and 20% triggered inhibition (pattern #2). In contrast, only 20% of thalamic neurons associated with post-spiking cortical excitation, while 80% of were associated with cortical inhibition patterns (45% with pattern #2% and 35% with pattern #3). Notably, pattern #3 was only identified with subcortical neuron spiking. Neither cortical cell depth, subcortical cell location (e.g. VPM vs VPL), nor cell-type classification (inhibitory and excitatory types [*Connors and Gutnick, 1990*; *Nowak et al., 2005*]) revealed any significant correlations between temporal dynamic clusters and cell-classification (not shown). We suggest future work with larger datasets and multiple cortical/subcortical areas could address the question of whether STMTD classification are a novel intrinsic single-cell property.

The profile of STMTDs across all mice and recordings were also presented by aligning their peak activation (red) or peak depression (blue) in various bilateral ROIs that span wide and varied regions of cortex (*Figure 8*). The analysis in *Figure 8* provides a powerful means of visualizing group-data temporal relationships across ROIs for both cortical and sub-cortically derived spikes. We found that as observed for individual neuron STMTDs, aggregate cortical neuron spikes were generally followed by strong periods of persistent activation (*Figure 8A*), whereas sub-cortical neurons were linked to more diverse cortical activity profiles, in particular longer depression across multiple cortical ROIs (*Figure 8B*).

## The role of movement and hemodynamics STMs

Our goal is to assess cortical functional connectivity based on coincidence between individual neuron spiking and ongoing spontaneous activity. However, we concede that in awake animals activity is rarely entirely spontaneous and that periods of volitional activity are present interspersed with relatively quiet intervals. Qualitative observations indicate limb twitches as well as tail and facial movements. In order to evaluate the impact of body movement on mapping, STAs were generated from spikes during periods of quietness and compared with STA from all spikes (*Figure 8—figure supplement 1*). Period of quietness and movement were identified by measuring behavior imaging collected simultaneously with the neurophysiological data (see Materials and methods, *Figure 8—figure supplement 1A* and *Video 5*). STA generated from periods of quietness did not differ from those generated with all spikes (see Materials and methods, *Figure 8—figure supplement 1B*). This suggests that, even when some movement could be observed during recordings, they minimally contribute to the mapping. This data also indicated that periods of high movement were relatively rare in awake head-fixed mice under the conditions we have employed and contributed negligibly to overall STA maps. Therefore, brain imaging activity obtained in awake states is mostly indicative of a quiet awake state and is not primarily movement-related activity.

Other potential sources of error include apparent changes in GCaMP6 signal due to alterations in blood volume or oxygenation (*Ma et al., 2016*; *Wekselblatt et al., 2016*). Increases in blood volume are expected to decrease both excitation and emission light. While we have implemented a multi-wavelength correction strategy (see Materials and methods), the corrections in practice did not appreciably alter cortical or thalamic STA maps or dynamics altering peak % $\Delta F/F_0$ by more than 1/10 of the maximum $\Delta F/F_0$. We provide examples of the strategy being employed (*Figure 7—figure supplement 1A,B*). In these examples, we employ a short blue reference signal that correlated positively with apparent blood volume artifacts that were revealed by parallel experiments using green reflected light imaging (*Figure 7—figure supplement 1C,D*). Consequently, the ratio of green over blue signal greatly reduced the blood volume hemodynamic response. To determine whether the short blue correction strategy was effective, we assessed data from GFP-m mice that exhibit green fluorescence signals that are not expected to be the calcium dependent as in GCaMP6 mice

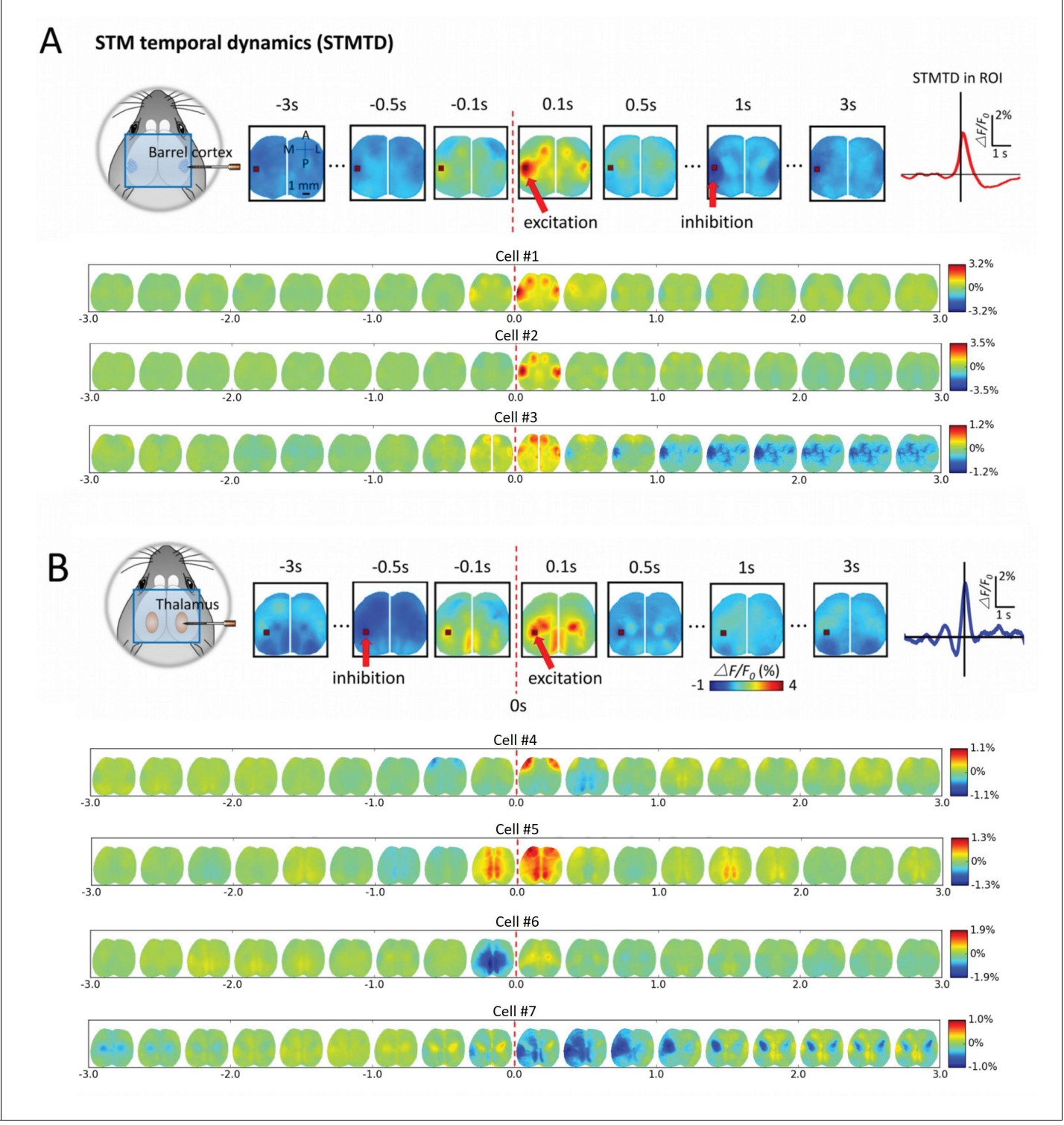

**Figure 6.** Montages of cortical and thalamic spatio-temporal dynamics. (**A**) Top: right hemisphere barrel cortex neuron spiking time montage stereotyped dynamics in left hemisphere barrel cortex region-of-interest (ROI). The maximally activated pixel in the ROI (red arrow) is tracked over time and reveals Spike-Triggered-Map Temporal Dynamics (STMTD) which rises quickly at spike time t = 0 and decays in 100–200 ms followed by 1–2 s cortical depression (red curve in right plot). Bottom: Additional examples of right hemisphere barrel cortex neuron spike-triggered montages show similar barrel-motor cortex activation pattern with peaks in cortical activation shortly following spiking and a return to baseline or prolonged depression. (**B**) Top: Same as in **A**, but for a thalamically recorded neuron (right hemisphere) which correlates strongly with motor cortex activation

*Figure 6 continued on next page*

*Figure 6 continued*

shortly after spiking. Bottom: Additional montage examples of thalamic neurons (also right hemisphere) reveal both the spatial diversity (i.e. different STMs) and temporal diversity (i.e. different STM dynamics).

The following figure supplements are available for figure 6:

**Figure supplement 1.** Single-cell STM and STMTDs are similar across spiking modes.

**Figure supplement 2.** Cortical cell STM and STMTDs are similar across spiking modes.

**Figure supplement 3.** Thalamic cell STM and STMTDs are similar across spiking modes.

**Figure supplement 4.** Single spike motifs sub-grouping reveals similar STMs across sub-networks.

**Figure supplement 5.** Additional examples of single cells STM stability in other mice and for larger partition sizes.

**Figure supplement 6.** Spike-triggered variance map.

(*Figure 7—figure supplement 1E*). We used the ratio of $F/F_0$ green fraction signal to blue reflected light signal $F/F_0$ fraction to reduce nonspecific signals that were observed in GFP mice. Consistent with published work (*Ma et al., 2016*; *Wekselblatt et al., 2016*) blood volume artifacts were greatly reduced. This same strategy was then applied to GCaMP fluorescence data. Although the approach was effective at removing smaller non-specific signals in GFP-m mice, in practice, using GCaMP mice, where much larger activity-dependent signals were present, we revealed only a relatively small apparent contribution of blood volume to cortical and subcortical STA fluorescent signals as in other work (*Vanni and Murphy, 2014*; *Murphy et al., 2016*; *Silasi et al., 2016*) (*Figure 7—figure supplement 1F,G*). Furthermore, thalamic STA maps and STA dynamic plots still indicated cases where thalamic spiking was associated with cortical inhibition. For these reasons, we have attempted to correct other STA maps using this multi-wavelength procedure.

## Discussion

Using STM of wide-field spontaneous calcium imaging data, we have characterized mesoscopic cortical maps defined by the spiking activity of individual cortical/sub-cortical neurons. Our results demonstrate that STMs can reveal functional cortical architecture related to the activity of individual cortical and subcortical-cortical neurons. Cortically recorded STMs reflect the cortical state when a neuron spikes in connected areas. We observed that the STMs of individual sub-cortical neurons had more variation than maps attributed to spiking cortical neurons. For example, sub-cortical STM patterns of neighboring neurons were more diverse than cortical neurons, and were less likely to match intra-cortical consensus activity patterns defined using SPM. Sub-cortical-neuron-derived STMs revealed multiple areas of activation and multimodal kinetic behavior, while intra-cortical spiking neuron networks were simpler in structure and kinetics. Furthermore, spiking sub-cortical neurons reflected diverse cortical multi-phasic excitation/inhibition timing patterns that were reflected in dynamic STMTDs. In contrast, most spiking cortical neurons were linked to a single phase of cortical excitation.

### Event-triggered brain mesoscale mapping

Previously, STA of local field potentials has been used to investigate of how single neurons in visual cortex were linked to on-going state-dependent activity (*Nauhaus et al., 2009*); however, this work only examined such correlations locally within visual cortex and did not assess regional connectivity using imaging or investigate differences with individual sub-cortical neurons. Other similar applications where single neuron spiking was recorded and related to spontaneous activity using calcium imaging have been restricted to in vitro brain slices (*Aaron and Yuste, 2006*). Our study extends previous in vivo work that assessed spike-triggered events using voltage-sensitive dye imaging (*Arieli et al., 1995*; *Tsodyks et al., 1999*) to encompass a more expansive spatial scale, electrode

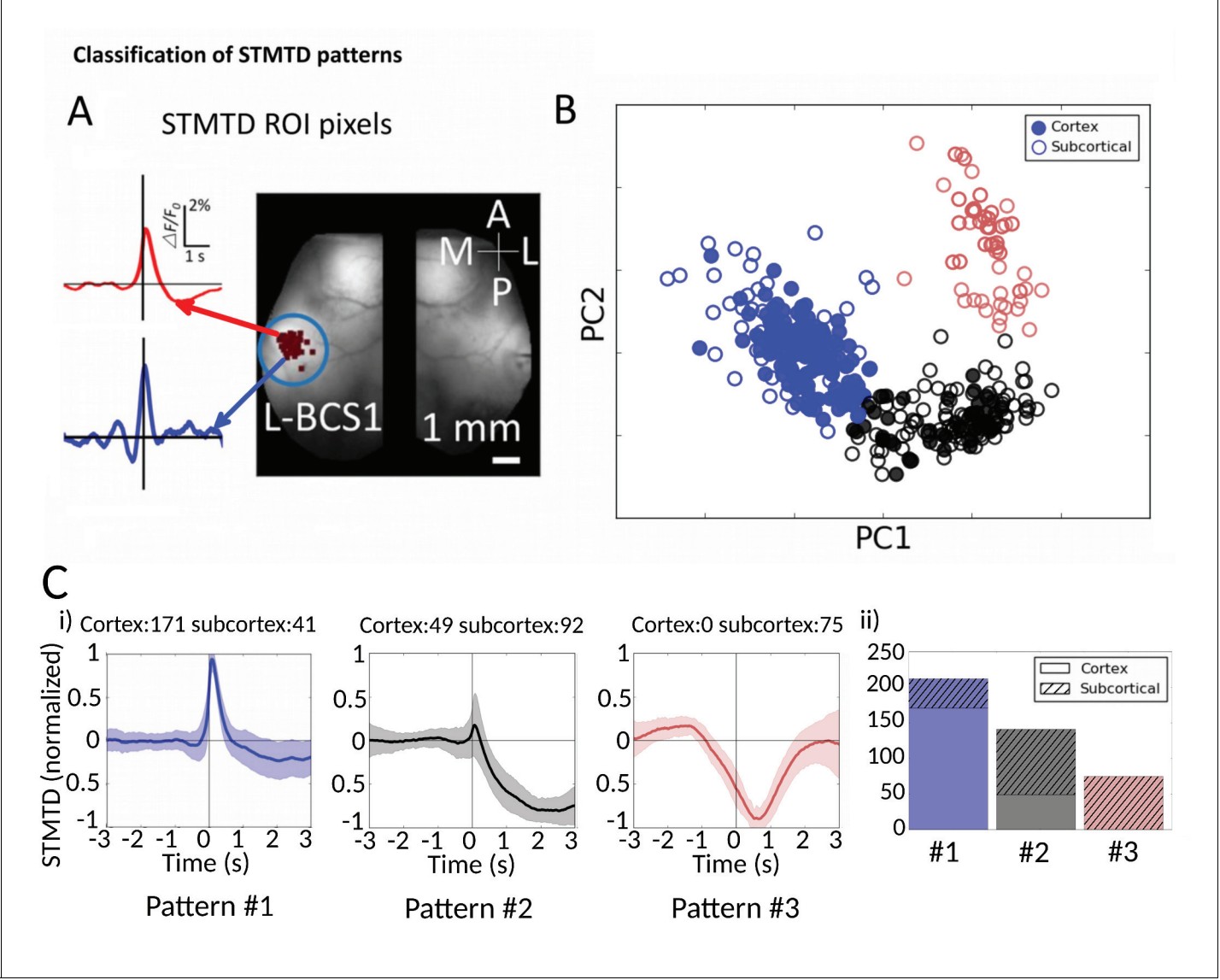

**Figure 7.** Classification of STMTD patterns. (**A**) Example of two STMTDs from pixels within L-BCS1 (left barrel cortex) from a single mouse recording (both cortical and subcortical neurons STMTD pixel locations shown). Each recorded cell STMTD has a slightly different maximum pixel amplitude location, but all fall within the L-BCS1 region. (**B**) STMTD PCA distribution from all 428 cortical and subcortical neurons recorded from all mice separated using KMEANS (k = 3). (**Ci**) STMTD patterns (±SD) classifications from (**B**). The number of neurons from cortex and thalamus used for the average are presented in title. (**ii**) Distribution of STMTD classification between cortical (clear) and subcortical (hashed) neuron generated STMTDs.

The following source data and figure supplement are available for figure 7:

**Source data 1.** Data files for PCA distribution clusters.

**Figure supplement 1.** Limited contribution of blood artifacts on STA.

arrays, awake recordings, and selective genetically encoded indicators of activity. While being important seminal findings (*Arieli et al., 1995*; *Tsodyks et al., 1999*), previous STA work was largely confined to the visual system, performed under anesthesia and was unable to define how collections of brain areas interact. The approach is probably most analogous to event-triggered MRI imaging from the standpoint of larger spatial scale (*Logothetis et al., 2012*). Interestingly, in this study, it was observed that during hippocampal ripple states that cortex exhibited net positive bold

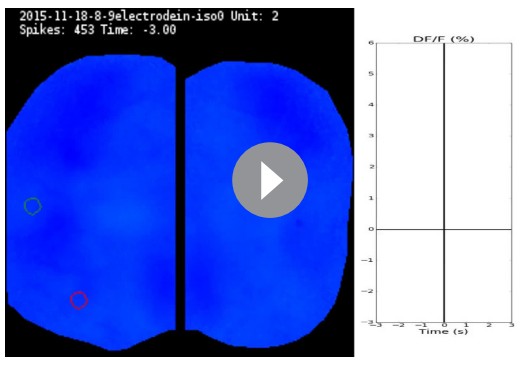

**Video 1.** Cortical neuron-triggered bilateral mesoscale calcium activity. Left: dorsal cortex calcium dynamics triggered from a right hemisphere barrel cortex recorded cell. Right, time-course reflecting dynamics in the left hemisphere barrel cortex (green) alongside a lower activated area (red). Dynamics in region-of-interest (green) exhibit high activation correlating with spiking time (t = 0 s) followed by a 2 s depression in fluorescence signal. Activity illustrated 3 s prior to, and 3 s following cell spiking.

responses and thalamus net negative BOLD responses. This anti-correlated relationship is consistent with some of our observations of thalamic spiking activity corresponding with cortical temporal dynamics exhibiting slow depression of calcium signals and may point to a larger coordinated network involving other brain structures. However, as we did not record simultaneously from hippocampus we cannot definitively comment on whether these observations necessarily correspond to hippocampal ripple events. Furthermore, MRI signals lack temporal resolution and can be more difficult to relate to neuronal activity than GCaMP signals that are isolated within excitatory neurons of GCaMP6f mice using specific promoters (*Chen et al., 2013b*; *Madisen et al., 2015*). Unique to our approach is the power to assess the functional connectivity and temporal dynamics between specific sub-cortical neurons and areas of cortex not predicted by previous knowledge such as linkages between thalamic neurons and cortical state as defined by GCaMP signal dynamics. We anticipate that the approach can be further refined as tools such as more selective cre-dependent CGaMP6f trans-

genic mice allow for the expression of calcium indicators in particular neuron types (*Madisen et al., 2015*). Furthermore, two-photon microscopy could be used to provide information about behavior of individual cells within the context of larger maps (*Chen et al., 2013a*; *Guo et al., 2014*; *Okun et al., 2015*). Because of the high sensitivity of the indicator and the possibility of measuring the activity of tens of single-units using multiple electrode channels simultaneously, a large number of functional connections can be mapped in only a few minutes of recording. Although we have only used a single electrode shank, we anticipate future applications using higher density electrodes and multiple shanks to collect spikes from more neurons simultaneously.

## Relationship between cortical and sub-cortical spiking derived maps

Our results indicate that the neocortex is divided into discrete subdivisions where individual

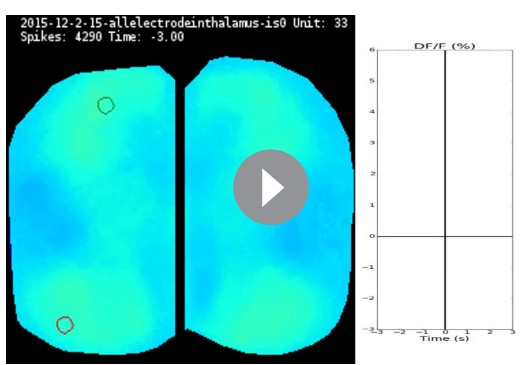

**Video 2.** Thalamic neuron #1 triggered bilateral mesoscale calcium activity. As in *Video 1* but from a right-hemisphere thalamic neuron. Region-of-interest (green) in the left hemisphere exhibits a transient pre-spiking depression followed by activation correlating with spike timing.

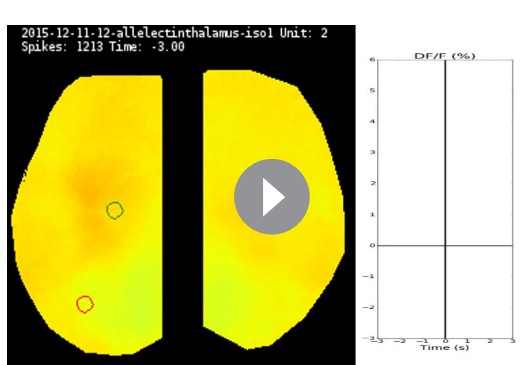

**Video 3.** Thalamic neuron #2 triggered bilateral mesoscale calcium activity. As in *Video 2* but from another thalamic neuron revealing peak activation in a different region-of-interest.

spiking cortical neurons generally belong to spatial-temporal maps that follow a consensus function that can be defined using correlation as in previous work (*Mohajerani et al., 2013*; *Chan et al., 2015*). In contrast, we show that single thalamic neurons tend to fire when cortex is in more kinetically diverse states which is dominated by inhibition. The more diverse dynamics between thalamic neurons and cortical mesoscopic networks indicate that sub-cortical thalamic neurons play an instructive role with respect to cortical state, particularly with respect to feedforward cortical inhibition (*Stroh et al., 2013*; *Urbain et al., 2015*), whereas cortical neurons may serve as relay endpoints or amplifiers (*Douglas et al., 1995*). A better understanding of these dynamics may yield insight into how disorders, such as epilepsy, and dementia, emerge when interactions between brain areas are disrupted (*Paz et al., 2013*; *Busche et al., 2015*;

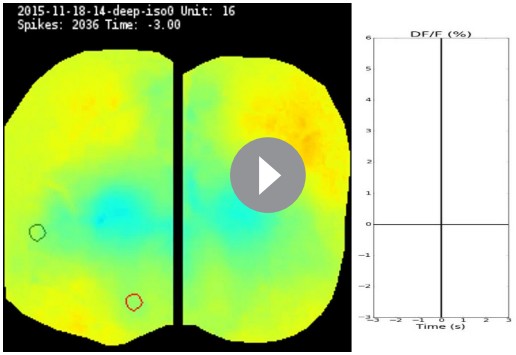

**Video 4.** Thalamic neuron #3 triggered bilateral mesoscale calcium activity. As in *Video 2*. Depressed cortical calcium activity is present in the left hemisphere barrel cortex region-of-interest prior to spiking and persists for an additional 2 s.

*McCormick et al., 2015*). The diversity in sub-cortical spiking derived maps may also reflect differing receptive field properties in thalamus and cortex based on varying types of functional convergence described previously (*Miller et al., 2001*). Indeed, in the somatosensory whisker barrel system, evidence for 'ensemble convergence' has been described where input from the thalamus can extend outside of the boundaries of the corresponding cortical receptive field (*Simons and Carvell, 1989*; *Linden and Schreiner, 2003*). The larger diversity of maps derived from the spiking of different thalamic neurons may be expected because of the smaller size of thalamic nuclei compared to the cortex and the recording of thalamic neurons from more-varied structures. Another potential source of variance may arise from the diversity of thalamocortical impulses that can be comprised of patterns of activity ranging from tonic, 'relay' transmission consisting of high regular rates of firing to burst-like activity where firing rates are low and interspersed with high-frequency events (*Steriade and Llinas, 1988*; *McCormick and Feeser, 1990*; *Sherman and Guillery, 1996*). Thalamic bursting can powerfully activate neocortical circuits and has been suggested to serve a 'wake-up' signal to sensory cortices (*Sherman and Guillery, 1996*; *Swadlow and Gusev, 2001*). When we segregated our recordings into various firing configurations, we did not observe profound differences in STMs or their temporal dynamics indicating that the averaging methodology is not constrained by a particular firing pattern. However, interpreting these results is caveated by the mesoscale resolution and calcium dynamics present in the recorded data.

## Applications of spike-triggered mapping

Mapping the functional connectivity around identified spiking neurons is important for understanding brain function and finding therapeutic targets for brain stimulation or brain machine interfaces. Identification of networks linked to individual neurons may help reveal the mechanism of brain machine interfaces where key signals are often attributed to only a small number of neurons (*Stanley et al., 1999*; *Serruya et al., 2002*; *Taylor et al., 2002*; *Guggenmos et al., 2013*). Other applications include understanding of how small groups of epileptic neurons (*Paz et al., 2013*) are coupled to brain networks leading to seizure propagation. Given that reciprocal connections between mesoscale structures are widespread, the cortical maps associated with a spiking neuron in a sub-cortical structure such as the sub-thalamic nucleus may provide clues as to how cortical activity can be manipulated to affect a sub-cortical target. This hypothesis can be tested by recording sub-cortically using electrode arrays, while stimulating regions of cortex that show coincident STMs using Channelrhodopsin-2 or other opsin-activity sensor pairs (*Lim et al., 2012*; *Rickgauer et al., 2014*; *Zou et al., 2014*; *Abdelfattah et al., 2016*; *Kim et al., 2016a*).

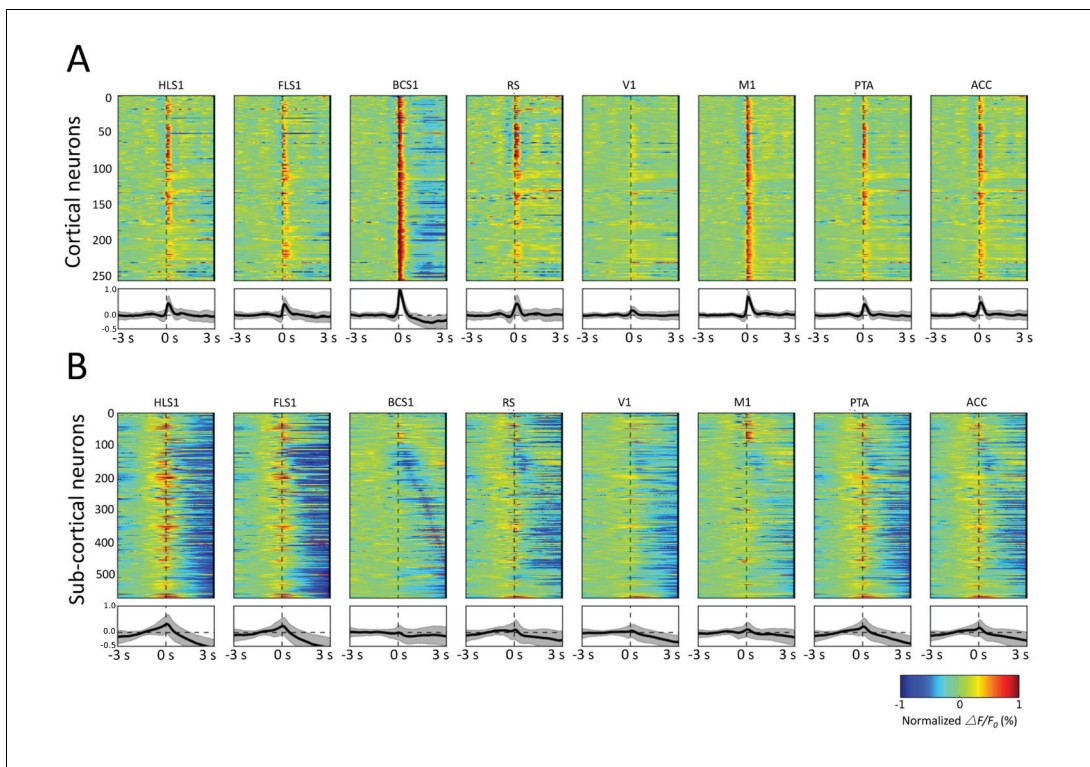

**Figure 8.** Spatiotemporal patterns of STMs. (**A**) Top: Normalized STMTDs (as in *Figure 7*) from maximum pixels tracked in multiple ROIs (HLS1, FLS1, BCS1, RS, V1, M1, PTA and ACC, see *Table 1*) for 255 cortical cells. Each horizontal line represents a single neuron's STMTD in each of the eight ROIs considered normalized to that neuron's STMTD maximum or minimum activation. Bottom: average and standard deviation of STMTD within each ROI for all cells. (**B**) Same as A, but for all thalamic neuron generated STMTDs. The thalamic STMTDs are more diverse, less temporally precise, and contain longer depression epochs – revealing ROI specificity and cortical vs. subcortical differences. These results are from awake mice.

The following figure supplement is available for figure 8:

**Figure supplement 1.** Limited contribution of body movement on STA.

## Kinetics of spike-triggered mapping

By computing spike-triggered calcium imaging averages, we reduce the contribution of neurons which fire out of phase. Analysis of STMTDs indicate slower time to peak (~100 ms) than postsynaptic potentials evoked by a single synaptic connection (time to peak ~20 ms) (*Bruno and Sakmann, 2006*). Slower dynamics are expected given the kinetics of GCaMP6 (*Chen et al., 2013b*). We have used deconvolution (*Pnevmatikakis et al., 2016*) to take into consideration the slower kinetics of GCaMP6 and compensate for it. Using this approach, we observed a significant acceleration of raw data but very modest effects on STMTD indicating that STMTD may already be accelerated relative to GCaMP6 kinetics by the statistical nature of spike/Ca$^{2+}$ transient temporal convergence. It is

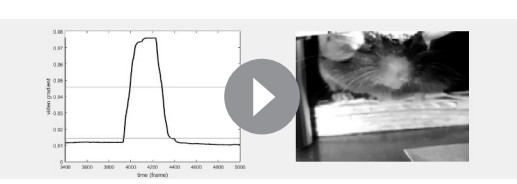

**Video 5.** Evaluation of whisker movement. *Left:* Average of absolute gradient within the region of interest (yellow box in *Figure 8—figure supplement 1*) between the frames 3400 and 5000. *Right:* Corresponding image frames displayed at real time.

also possible that slower dynamics of STMTD reflect sequences of spiking activity propagating through specific polysynaptic circuits. This speculation was supported by the similar time range of STMTD and cue-triggered recall of learned temporal sequences (*Xu et al., 2012*). STMTD kinetics can be improved using faster sensors such as organic voltage-sensitive dyes (*Shoham et al., 1999*; *Mohajerani et al., 2013*), or genetically encoded voltage (*Carandini et al., 2015*; *Gong et al., 2015*; *Abdelfattah et al., 2016*) or glutamate sensors (*Xie et al., 2016*).

### Extension to behaviorally driven activity

We acknowledge that the same approach can be extended to making STMs during specific behaviors. During specific behaviors we do not expect major shifts in area map boundaries found during spontaneous activity, as we believe these are largely determined by projection anatomy (*Mohajerani et al., 2013*; *O'Connor et al., 2013*; *Oh et al., 2014*; *Zingg et al., 2014*) and in the case of sub-cortical neuron maps (HPF for example) poly-synaptic, hard-wired connections. During behavior we expect more nuanced changes in the weighting, timing, and frequency-dependence of STM networks during an active task. It is possible that specific behaviors will reveal the superposition of multiple cortical motifs associated with progression through the task. STM mapping of cortex would be particularly interesting the context of rhythmic whisking-related centers within the medulla and thalamus and their linkage to cortical maps within barrel-motor areas (*Moore et al., 2013*; *Deschênes et al., 2016*; *Sreenivasan and Petersen, 2016*).

### Conclusion

We conclude that the analysis of single neuron spiking activity can reliably reflect mesoscale activity transitions within mouse cortex. STMs together with connectomic information (*Hunnicutt et al., 2014*; *Oh et al., 2014*; *Zingg et al., 2014*; *DeNardo et al., 2015*) may help to bridge the gap between single neuron function and larger networks. Our data have already revealed that thalamic neurons interact with cortex during specific state transitions that not reflected by typical consensus cortical neuron behavior. We have exploited spontaneous activity as a means of sampling active cortical networks. The presence of such obligate long-range relationships in even spontaneous activity may suggest new opportunities and routes by which brain stimulation and inhibition can be applied to affect synaptically connected areas.

## Materials and methods

### Animals

Animal protocols (A13–0336 and A14–0266) were approved by the University of British Columbia Animal Care Committee and conformed to the Canadian Council on Animal Care and Use guidelines and animals were housed in a vivarium on a 12 hr day light cycle (7 AM lights on). Most experiments were performed toward the end of the mouse light cycle. Transgenic GCaMP6f mice (males, 2–4 months of age, weighing 20–30 g; n = 16), were produced by crossing Emx1-cre (B6.129S2-$Emx1^{tm1(cre)Krj}$/J, Jax #005628), CaMK2-tTA (B6.Cg-Tg(Camk2a-tTA)1Mmay/DboJ, Jax #007004) and TITL-GCaMP6f (Ai93; B6;129S6-$Igs7^{tm93.1(tetO-GCaMP6f)Hze}$/J, Jax #024103) strain (*Madisen et al., 2015*). Transgenic GCaMP6s mice (n = 3) were produced by crossing Emx1-cre (B6.129S2-$Emx1^{tm1(cre)Krj}$/J, Jax #005628), CaMK2-tTA (B6.Cg-Tg(Camk2a-tTA)1Mmay/DboJ, Jax #007004) and TITL-GCaMP6s (Ai94;B6.$Cg$-$Igs7^{tm94.1(tetO-GCaMP6s)Hze}$/J, Jax #024104) strain. Transgenic GCaMP3 mice (n = 8) were produced by crossing Emx1-cre and R26-GCaMP3 (Ai38; B6;129S-$Gt(ROSA)26Sor^{tm38(CAG-GCaMP3)Hze}$/J, Jax #014538) strain (*Zariwala et al., 2012*; *Vanni and Murphy, 2014*). The presence of GCaMP expression was determined by genotyping each animal before each surgical procedure with PCR amplification. These crossings are expected to produce a stable expression of the three calcium indicator variants (GCaMP3, GCaMP6s and GCaMP6f [*Chen et al., 2013b*]) specifically within all excitatory neurons across all layers of the cortex (*Vanni and Murphy, 2014*). Control experiments, assessing the specificity of STM, were performed in Thy1-GFP-M mice (n = 6; Jax #007788). No method of randomization was used since all mice belonged to the same sample group. Samples sizes were chosen based on previous studies using similar approaches (*Mohajerani et al., 2013*; *Vanni and Murphy, 2014*; *Chan et al., 2015*). Given the use of automated acquisition and analysis procedures we did not employ blinding.

## Surgery

Mice were anesthetized with isoflurane (1.5–2%) for induction and during surgery and a reduced maintenance concentration of isoflurane (0.5–1.0%) or urethane was used later during anesthetized data collection. In some cases, animals were allowed to wake up following isoflurane anesthesia for awake imaging (see section 'Multimodal recording in awake mice'). Throughout surgery and imaging, body temperature was maintained at 37°C using a heating pad with a feedback thermistor. For cortical experiments, mice were placed on a metal plate that was mounted on a macroscope. The skull was fastened to a stainless steel head-plate and was connected with tubing to a water pump, which circulated temperature controlled 37°C water to ensure physiological temperature. A 9 × 9 mm bilateral craniotomy (bregma 3.5 to −5.5 mm, lateral −4.5 to 4.5 mm) covering multiple cortical areas was made as described previously (*Mohajerani et al., 2013*). For sub-cortical experiments, mice were placed in a stereotaxic apparatus and an incision was made in the midline to expose the skull as in cortical experiments. A burr hole was then unilaterally drilled (usually in the right hemisphere) above the thalamic area (stereotaxic coordinates considering a 45° angle (*Figure 1A*): between 1.7 ± 0.3 mm posterior to bregma and 1.6 ± 0.4 mm lateral to midline. We estimated angular tilt relative to a perpendicular penetration to the cortical surface of less than 5° (*Hunnicutt et al., 2014*). In order to minimize movement artifact (due to breathing and heartbeat), the exposed skull was fastened to a stainless steel head-plate with cyanoacrylate glue and dental cement. In cases where the laminar probe was inserted (as opposed to a glass electrode), a craniotomy was only made for the probe insertion site and cortical GCaMP imaging was performed through intact bone.

## In vivo single unit recording

For initial glass pipette recordings, the pipette was advanced into the targets (HLS1, FLS1, BCS1, V1, M1, ACC and RS) on the cortical surface (within the right hemisphere) at an angle of 30° from the horizontal using a motorized micromanipulator (MP-225, Sutter Instrument Company). Signals were recorded through a silver wire placed inside the micropipette and were amplified (MultiClamp 700A or Axopatch 200B, Molecular Devices) and digitized at 12.5 kHz (Digidata 1322A, 16-bit Data Acquisition System). A reference electrode, teflon-coated, chlorided silver wire (0.125 mm) was placed on the right edge of the craniotomy. For multichannel 16-channel laminar electrode (Neuro-Nexus, A16–10 mm-100-177) recordings, the electrode was directed toward the center of the burr hole using a motorized micromanipulator (MP-225, Sutter Instrument Company). The electrode was inserted into the right hemisphere with a 45° angle from the lateral surface of the cortex to avoid contact with the region where wide-field imaging was performed. The electrode was first held in cortex (BCS1) for cortical recordings and then advanced into the sub-cortical areas. The multichannel signal was amplified using 16-channel data acquisition system (20 kHz, USB-ME16-FAI-System, Multi-Channel Systems) and recorded for at least 5 min for each recording site. To minimize tissue damage, all experiments were performed by only a single insertion of the laminar electrode, and most trajectories were from barrel cortex to thalamus.

## Calcium imaging

Images of the cortical surface were recorded through a pair of front-to-front video lenses (50 mm, 1.4 f:30 mm, 2 f) coupled to a 1M60 Pantera CCD camera (Dalsa) (*Vanni and Murphy, 2014*). To visualize the cortex, the surface of the brain was illuminated with green light (but not during image acquisition). Calcium indicators were excited with blue-light-emitting diodes (Luxeon, 470 nm) with bandpass filters (467–499 nm). Emission fluorescence was filtered using a 510–550 nm bandpass filter or collected in a multi-band mode as described below. For single wavelength green epifluorescence, we collected 12-bit images at varying time resolution (20–100 ms; i.e., 10–50 Hz) using XCAP imaging software. In order to reduce file size and minimize the power of excitation light used, we typically bin camera pixels (8 × 8) thus producing a resolution of 68 μm/pixel. These imaging parameters have been used previously for voltage-sensitive dye imaging (*Mohajerani et al., 2013*) as well as anesthetized GCaMP3 imaging of spontaneous activity in mouse cortex (*Vanni and Murphy, 2014*) and awake GCaMP6 imaging in mouse cortex with chronic window (*Silasi et al., 2016*).

In some experiments (as indicated), we employed a multi-wavelength strategy to correct for potential green epifluorescence signals that were associated with non-calcium dependent events. We employed a variant of the elegant spectral correction strategy described by others (*Ma et al.,*

2016; *Wekselblatt et al., 2016*) that monitor changes in green reflected light near the isobestic point of hemoglobin. This strategy was inspired by previous work using blue excitation/reflected light(*Sirotin, 2010*). We assume hemoglobin is the primary absorber in brain tissue in vivo and changes in blood volume or oxygenation affect both excitation and emission of light used for wide-field imaging (*Ma et al., 2016*). Our strategy makes use of short blue wavelength reference light that is also near a hemoglobin isosbestic point. While others have used a strobed LED presentation with a subset of frames providing a green reflected light reference image (*Ma et al., 2016*; *Wekselblatt et al., 2016*), we took advantage of an RGB camera sensor to allow simultaneous acquisition of a shorter wavelength blue ~447 nm signal that correlates strongly with green reflected light signals. This strategy provides a short blue light reference without the need for strobing which can limit time resolution and potentially entrain some neuronal rhythms (*Iaccarino et al., 2016*) and is more technically demanding from a hardware synchronization standpoint. Our strategy employs the Raspberry Picam's RGB sensor (Waveshare Electronics RPi Camera F) to independently resolve signals attributed to blood volume changes as blue reflected light, while simultaneously collecting green epi fluorescence (GCaMP6). We used a Chroma 69013m multi-band filter 10 mm diameter mounted just over the image sensor allowing blue, green, and red signals to be simultaneously obtained in separate channels of the cameras RGB sensor with less than 10% cross talk between channels. We employ 2 Luxeon LEDs: (1) Royal-Blue (447.5 nm) LUXEON Rebel ES LED with added Brightline Semrock 438/24 nm filter to provide a short blue wavelength reflected light signal that is expected to report blood volume changes; (2) a blue 473 nm Luxeon Rebel ES LED for excitation of GCaMP6 with a Chroma 480 nm/30 nm excitation filter. In preliminary analysis, we found that the short blue signal correlated positively with apparent blood volume artifacts that were revealed by parallel experiments using green reflected light imaging (r = 0.93, see *Figure 7—figure supplement 1C*). Given that the short-blue reflected light signals provided a surrogate indicator of green reflected light (they are highly positively correlated) we used this in a ratiometric correction strategy. While the shorter blue wavelength light will scatter more than a green strobed reflected light signal used by others (*Ma et al., 2016*; *Wekselblatt et al., 2016*), our analysis of green reflected light and short blue reflected light, indicating that two were highly correlated, suggests that the major artifacts we observe are associated with large blood volume changes in superficial cortical layers.

## Multimodal recording in awake mice

To initiate wakefulness isoflurane and oxygen were stopped and the anesthesia mask was removed. Calcium imaging data were obtained over the following 1 hr. The body temperature of mice was maintained with a heating pad. Awake calcium imaging of spontaneous activity was performed in the absence of visual and auditory stimulation. A behavioral monitoring camera was used to confirm that the mice were indeed awake and relatively unstressed as grooming and whisking were occasionally observed. An analgesic, buprenorphine, was injected (0.075 mg per kg body weight intraperitoneally) 24 hr before awake calcium recordings. We used a second Dalsa 1M60 camera (150 Hz) or Raspberry Picam's RGB sensor (60 Hz) to capture body and whisker movements under infrared illumination.

While relatively few large body movements were observed during awake imaging sessions, their impact on mapping was evaluated by generating STA from period of quietness. To identify regions of movement or quietness, the standard deviation of luminance fluctuation was calculated for each pixel. This approach showed that most of the movements were localized on the facial (whisker and jaw) and forepaw regions. A region of interest was manually drawn for each frame and the sum of absolute value of the gradient was calculated by subtracting each frame by their previous within this region. This gradient profile within the region of interest was temporally smoothed at 0.1 Hz, and the median and standard deviation were calculated (σ). Periods of quietness were identified as having a gradient lower than [median+σ/10], while periods of movement were higher than [median+σ]. To more selectively identify periods of quietness isolated from any movement, an exclusion window of 10 s was applied at the beginning and the end of each period of quietness and only periods of more than 10 s were kept. STA was then generated only from spikes for periods of quietness and compared with STA of all spikes.

## Sensory stimulation

Sensory stimuli were used to confirm sensory cortical and sub-cortical areas using forelimb, hindlimb, whisker and visual stimulation. To stimulate the forelimbs and hindlimbs, thin acupuncture needles (0.14 mm) were inserted into the paws, and a 0.2–1 mA, 1 ms electrical pulse was delivered. To stimulate a single whisker (C2), the whisker was attached to a piezoelectric device (Q220-A4-203YB, Piezo Systems, Inc., Woburn, MA) and given a single 1 ms tap using a square pulse. The whisker was moved at most 90 μm in an anterior-to-posterior direction, which corresponds to a 2.6° angle of deflection. A 1 ms pulse of combined green and blue light was delivered as visual stimulation. Averages of sensory stimulation were calculated from 20 to 40 trials of stimulation with an inter-stimulus interval of 10s.

## Single unit activity analysis

Raw extracellular traces were imported into Spike2 (Cambridge Electronic Design, Cambridge, UK) or SpikeSorter software (*Swindale and Spacek, 2014*) for spike sorting and analysis. Briefly, data were high pass-filtered at 1 kHz, and single spikes were detected using a threshold of 4.5 times the median of the standard deviation over 0.675. Sorting was carried out by an automated method previously described (*Swindale and Spacek, 2014*) and followed by manual visual inspection of units. For analysis, we only used only units with a peak-to-peak extracellular amplitude of at least 40 μV, a minimum of 200 spikes, and a calcium cortical response (STMTD, see next section) of at least 0.5% to improve single unit sorting isolation, STM map stability and signal-to-noise ratios for STMTD clustering (see main text), respectively.

## Calcium imaging analysis

TIF files of raw fluorescence were imported and processed using Matlab (Mathworks, Inc. Natick, MA) or Python2.7 custom codes. For each spike, we averaged the cortical fluorescence of every pixel during the period preceding the spike ($-3$ to 0 s: baseline). $\Delta F/F_0$ was then performed by subtracting and dividing the baseline to each frame for each spike within a time window of $\pm 3$ s. We tested additional methods for computing $\Delta F/F_0$ including computing baseline from the average of the entire recording or band pass filtering the data (0.1 Hz to 6.0 Hz) but the results were similar. STA (spike-triggered average) sequences were created by averaging the $\Delta F/F_0$ responses for each trial and were compared with random spike STA. STA maps (STM) were then defined as the maximum response calculated for each pixel within a time window of $\pm 1$ s.

STM temporal dynamics (STMTDs) were defined as the time course of activity of the maximally activated pixel in a region of interest. The resulting clusters were separated using k-means clustering algorithm. Putative cell classification into inhibitory and excitatory cell types was based on the full-width-half-max of each unit's positive and negative phases (*Connors and Gutnick, 1990*; *Pape and McCormick, 1995*).

To calculate seed pixel correlation maps (SPM), the contribution of global and illumination fluctuations was subtracted from the signal of each pixel (*Vanni and Murphy, 2014*) and the spontaneous activity recording sequences were temporally pass-band filtered (0.3–3 Hz). Then, cross-correlation coefficient r values between the temporal profiles of one selected pixel and all the others were calculated (*White et al., 2011*; *Mohajerani et al., 2013*; *Vanni and Murphy, 2014*). Similarities between STM and SPM maps were performed by measuring the r-value Pearson correlation coefficient between each pair of pixels.

To compare the STM with anatomical database, brain stacks of 140 slices were downloaded from the Allen Mouse Brain Connectivity Atlas providing AAV-virus tracing database (http://connectivity.brain-map.org/, [*Oh et al., 2014*]). For each slice, the first dorsal 300 μm of brain fluorescence in the Z-axis were summed to generate partial maximum Z-projection maps similarly to previous studies (*Mohajerani et al., 2013*).

## Deconvolution

We temporally deconvolved our calcium imaging pixel-by-pixel using the method presented by *Pnevmatikakis et al. (2016)* and code provided by the authors on Github (https://github.com/epnev/ca_source_extraction). Briefly, the method uses an autoregressive approach to estimate the calcium transient as an impulse response from the data itself (*Pnevmatikakis et al., 2016*). Making

use of this, the time course of calcium transients is then deconvolved using a computationally efficient non-negative, sparse, constrained deconvolution.

## Cell spiking mode determination

We implemented methodology previously described for defining main spiking modes (*Sherman et al., 2006*) (Figure 6.5, pg 236). Briefly, the method requires determination of the distribution of each spike's inter-spike-interval (ISI) between the previous (x-axis) and following (y-axis) spike (*Figure 6—figure supplement 1*). The distribution is then plotted using logarithmic scales and naturally arising clusters are grouped or clustered. Spike groups occurring in approximately each quadrant of the plot indicate different spiking modes: first spikes in a burst (bottom right), spiking occurring during a burst (bottom left), last spikes in a burst (top left), and tonic spikes (top right). The vast majority of cortical cells we recorded in barrel cortex did not exhibit multiple classes of spiking modes (2 examples provided where some natural clustering is present: *Figure 6—figure supplement 1A and B*), and only a few of thalamic cells we recorded showed clear bursting modes (two-examples provided *Figure 6—figure supplement 1C and D*) while also passing our minimum thresholds (see Materials and methods: **Single unit activity analysis**).

## Single-spike motif sub-network analysis

We sought to determine whether sub-groups of spike STMs from a single cell could cluster and yield average STMs that were different than the all-spike average STM. Our approach was to group single spike motifs by similarity in a high-dimensional space (e.g. 64 × 64 pixels=4096 dimensions) and removed spontaneous motifs averages that were similar to our groupings to reveal the underlying single cell STM. The first step was to compute distributions of the cortical STMs of all spikes (which have high variability; *Figure 6—figure supplement 4A*) in a high dimensional space. The lack of obvious clusters indicated that there were no sub-groups of STMs present in the data which are removed by the averaging procedure. We proceeded to group the STM distributions by similarity into four (or more) partitions to reveal active sub-networks present during single spiking (*Figure 6—figure supplement 4B*). While inter-spike-interval (ISI) distribution were largely similar for the sub-grouped networks (*Figure 6—figure supplement 4C*) the resulting four sub-networks had substantial diversity indicating that (on averae) spiking occurred during different types of active cortical networks with only one of these sub-networks resembling our all-spike average STM (*Figure 6—figure supplement 4D*; four sub-network STMs and sum at the bottom). We next identified spontaneous STMs – that is STMs occurring without spiking – that were most similar to our sub-networks to subtract their contribution and reveal the cell's component STM in the sub-network STMs. We thus grouped spontaneously occurring motifs (*Figure 6—figure supplement 4E*) into sub-networks similar to spike triggered networks by re-using the spike generated sub-network centres (*Figure 6—figure supplement 4B,F*). This ensured that the spontaneous sub-networks would be similar to the spike-triggered sub-networks. The resulting spontaneous sub-networks are similar – but not identical – to the cell spike triggered sub-networks (*Figure 6—figure supplement 4H*; note that the sum is mostly noise as expected when summing over all activity). Importantly, when subtracting the spontaneously active sub-network STMs (*Figure 6—figure supplement 4H*) from the spike-triggered sub-network STMs (*Figure 6—figure supplement 4D*), we recovered STMs which represented mainly single cell spiking components and were largely similar to the overall average STM. We provide additional examples of this STM partitioning using other cortical and thalamic cells and also an example where we partitioned the STM distribution into 12 sub-networks but still recovered the all-spike average STM from each of the partitioned networks (*Figure 6—figure supplement 5*).

## Code availability

A mixture of custom Python and Matlab code was used for analysis (https://github.com/catubc/sta_maps).

## Histology

Prior to each recording, pipettes were filled with dye (Texas red-dextran) or the rear of a laminar electrode shank (side opposite the recording sites) was painted with fluorescent 1, 1-dioctadecyl-3,3,3,3-tetramethylindocarbocyanine perchlorate (DiI, ~10% in dimethylfuran, Molecular Probes,

Eugene, OR). Dye-labeled pipettes and electrodes were not used until the dimethylfuran solvent had evaporated. At the end of each experiment, animals were killed with an intraperitoneal injection of pentobarbital (24 mg). Mice were transcardially perfused with PBS followed by chilled 4% PFA in PBS. Coronal brain sections (50 µm thickness) were cut on a vibratome (Leica VT1000S). Images of diI labeling with counter-stained DAPI were acquired using confocal microscopy (Zeiss LSM510) to reveal the electrode track and help identify the approximate subcortical location of recorded single units.

## Statistics

Data were analyzed using GraphPad Prism six and custom written software in MATLAB and Python2.7. Mann-Whitney non-parametric tests were used to compare correlation coefficients between STMs, percentage overlap between STM pairs, and the correlation coefficients between STMs and SPMs. **** denote $p < 0.0001$. Collection of data and analysis were not performed blind to the nature of the experiment, and there was no randomization of animal groups. STMs included in analyses were generated from a minimum of 200 spikes. Only STMTDs exhibiting fluorescence exceeding 0.5% $\triangle F/F_0$ were used for analysis. Sample sizes were not pre-determined but are consistent with previous experiments using similar methodology (*Mohajerani et al., 2013*; *Vanni and Murphy, 2014*; *Chan et al., 2015*).

# Acknowledgements

This work was supported by Canadian Institutes of Health Research (CIHR) Operating Grant MOP-12675 and Foundation Grant FDN-143209 to THM, THM is supported by the Brain Canada Neurophotonics Platform, an International Alliance of Translational Neuroscience (IATN) program to DX. We thank Pumin Wang, Cindy Jiang for surgical assistance; Jamie D Boyd and Federico Bolanos for technical assistance and Alexander McGirr for helpful discussions and comments on the manuscript. We thank Hongkui Zeng and Allen Brain Institute for providing transgenic mice.

# Additional information

## Funding

| Funder | Grant reference number | Author |
|---|---|---|
| International Alliance of Translational Neuroscience | | Dongsheng Xiao |
| Brain Canada, Canadian Neurophotonics Platform | | Jeffrey M LeDue Timothy H Murphy |
| Canadian Institutes of Health Research | MOP-12675 | Timothy H Murphy |
| Canadian Institutes of Health Research | FDN-143209 | Timothy H Murphy |

The funders had no role in study design, data collection and interpretation, or the decision to submit the work for publication.

## Author contributions

DX, Conceptualization, Data curation, Formal analysis, Investigation, Methodology, Writing—review and editing; MPV, Conceptualization, Software (Matlab), Formal analysis, Investigation, Methodology, Writing—review and editing; CCM, Conceptualization, Software (Python), Formal analysis, Investigation, Methodology, Writing—review and editing; AWC, Conceptualization, Software, Formal analysis, Investigation, Methodology, Writing—review and editing; JML, Conceptualization, Formal analysis, Methodology, Writing—review and editing; YX, Conceptualization, Investigation; ACNC, Conceptualization, Supervision, Methodology; NVS, Conceptualization, Resources, Software, Methodology, Writing—review and editing; THM, Conceptualization, Supervision, Funding acquisition, Validation, Investigation, Methodology, Writing—original draft, Project administration, Writing—review and editing

## Author ORCIDs

Timothy H Murphy, http://orcid.org/0000-0002-0093-4490

## Ethics

Animal experimentation: Animal protocols (A13-0336 and A14-0266) were approved by the University of British Columbia Animal Care Committee and conformed to the Canadian Council on Animal Care and Use guidelines.

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
