## [Decision Letter]

Thank you for submitting your article "Mapping cortical mesoscopic networks of single spiking cortical or sub-cortical neurons" for consideration by *eLife*. Your article has been reviewed by two peer reviewers, and the evaluation has been overseen by David Kleinfeld as Reviewing Editor and Timothy Behrens as the Senior Editor. The reviewers have opted to remain anonymous.

The reviewers have discussed the reviews with one another and the Reviewing Editor has drafted this decision to help you prepare a revised submission.

Summary

The authors combined single neuron electrical recordings of spiking output with wide field imaging of the calcium transients in cortex. This allows them to form a transfer function of the spatiotemporal patterns in cortex in response to spiking from a single neuron in with cortex or thalamus. These patterns may extend across multiple cortical areas, consistent with long-range connections. Further, they are more robust when derived with respect to cortical versus thalamic spikes. These data provide key, additional evidence for a highly restricted number of patterns generated by cortex, even from "spontaneous" activity, compared with the number of neurons in a region of interest.

Essential revisions:

Both reviewers as well as the Reviewing editor find this to be a carefully executed set of experiments and praise to quality of the imaging over an unprecedented nearly 1 cm by 1 cm field of view. The reviewers raise excellent issues and I would like the author to respond to all comments. Regarding "essential" points, one reoccurring theme that will require additional analysis concerns the averaging. First, it would be very useful to see variance maps as well as average or spike triggered maps (STMs) as an extension of Figure 3 and Figure 4. The two may not be scaled versions of each other if the calcium responses are nonlinear. For that reason, the authors may further consider an analysis of the covariance of the data, as opposed to only the mean, as would reveal properties of "complex" versus "simple" visual cells. Perhaps this issue is at bay with the thalamic data. An example of how to do this was reviewed in Aljadef et al. (Neuron 2016), where the calcium images would replace the visual stimuli in the analysis.

The reviewers raise three additional points that must be considered and preferably implemented. The first is the important issue of removing potential blood flow contributions, as was done by Chris Niell in a 2016 Journal of Neurophysiology manuscript. This should, ideally, be applied to the present data if at all possible. The second concerns the nature of the STM calculated from temporally isolated spikes versus spikes that are part of a burst. Given the known depression of thalamo-cortical synapses, this could likely contribute the variability of the STM data using thalamic spikes. To address this issue, the thalamic bursts can be isolated from the spike records and, as one possibility, the STM can be computed with respect to only the first spike. Lastly, the "spontaneous activity" may currently include sensory-evoked activity from body motion. As the authors recorded body motion, it would be germane to check if "spontaneous activity" in fact results from self motion, as this changes the interpretation of "spontaneous activity' from internally generated to reafferent driven.

Reviewer #1:

Spontaneous activity recorded with various methods (e.g., fMRI, intrinsic signal, voltage, and Ca^2+^ imaging) in the brain is useful to predict anatomical and functional connectivity in the brain. For example, seed pixel correlation maps (SPMs) can show the cross-correlation map of the spontaneous activity for a defined seed pixel in the images. In this study, Xiao et al. combined GCaMP imaging of an entire cortex with single neuron electrophysiological recordings, and used the spike-triggered averaging approach to determine the mesoscopic network motif linked to the recorded neurons. The spike trigger averaging approach has already been reported by Grinvald lab (Arieli et al., 1995), and the current study does not provide a new methodological framework. However, combined with state-of-the-art GCaMP imaging, Xiao et al. nicely demonstrates the power of this approach by finding subcortical to cortical network motifs, which have been difficult with optical imaging alone.

Overall, this study is technically sound. I suggest the following points to improve the quality and/or the presentation of the work.

Major issues:

1) Results section. Spike-trigger averaging of optical signals was originally reported by Grinvald lab and authors are simply adopting this approach to the large-scale GCaMP imaging. Authors need to introduce the original work in the Introduction and/or Results section.

2) Figure 1. Authors are using single-spike events to calculate the STM. However, in GCaMP imaging, single spikes can produce only marginal response changes (ΔF/F). In addition, a single spike in a single neuron is unlikely to effectively activate all the post-synaptic neurons. Thus, it is possible that multiple-spike events (or bursts) can produce a more reliable STM. The authors could test this possibility by reanalyzing data and the results could be shown as in Figure 2.

3) Figure 2. Authors demonstrate that a conserved network motif emerges by averaging the spike-triggered maps. However, this is based on an assumption that a single spike is always linked to a single type of network motif. Is it possible that multiple motifs are linked to a single neuron and authors' approach is averaging out multiple types of motifs? Authors could, for example, try classification of the spike-triggered maps.

4) Figure 3. Given the diversity of cell types (and projection patterns) within a defined brain region, it is possible that STM is diverse even within a defined cortical area. Did authors find such cases? (In fact, STMTD is diverse in Figure 7 and Figure 8.) It is also possible that neurons in different cortical layers show distinct STMs even within the same area. Throughout this manuscript, box plots or beeswarm plots may be more appropriate if they show non-normal distributions with rare examples.

Reviewer #3:

Neuroscience research linking ongoing activity in brain-wide networks to that of single neurons is sorely lacking. fMRI studies have revealed interesting default brain networks that correspond to different behavioral states. The current study analyzes such patterns using widefield calcium imaging, selectively in excitatory neurons throughout dorsal cortex – arguably a useful complementary approach to the aforementioned human work given that calcium signals are more tightly linked to spiking activity. In addition, previous work using hippocampal electrophysiology and fMRI in non-human primates (Logothetis et al., Nature 2012) showed that hippocampal ripple activity during states other than active waking was linked to widespread cortical activation as well as to suppressed activity in subcortical regions (including thalamus), but few studies have taken a similar approach using widefield calcium imaging. The current study uses impressive tools, the combination of which is somewhat novel, to link spiking activity from single cortical and subcortical neurons with widespread GCaMP6 calcium activity during anesthesia and quiet waking. Further, only few studies have performed imaging with such a large field of view (9 mm x 9 mm). In general, the manuscript is well written. The successful collection of these datasets is exciting, and the combination of dense imaging, electrophysiology and behavioral information provides an extremely rich and unique dataset. However, the analyses performed do not effectively extract much of this richness, and do not, in the current form, reveal clearly interpretable insights of broad interest gained by this approach (particularly given the caveats involved in averaging across many neurons). Further, several interpretations of the data are confusing, and some key references require further consideration. These issues are discussed in detail below:

1) The authors show that seed pixel correlation maps (SPMs) are better matched to cortical single-unit spike-triggered maps (STMs) than to thalamic STMs, and that thalamic STMs are generally more diverse than cortical (specifically barrel cortex) STMs. However, as suggested by the authors in the Discussion, this is likely due, at least in part, to greater diversity across thalamic recordings from neurons in a larger number of smaller thalamic nuclei (vs. more consistently from S1 barrel region of cortex, albeit across layers) vs. recordings in a single cortical area with a large magnification factor. Further, cortical neurons within S1 are highly correlated, likely more so that nearby neurons in thalamus, further ensuring strong correlation between the cortical spiking activity and the SPM maps (since the seed cortical pixel is likely highly correlated with the cortical single-unit recording), more so that would be likely be possible with most thalamic units). The other suggested explanations for these findings made by the authors in the Discussion are far more speculative, and many would be difficult to test. Further, little effort is made to relate specific thalamic recordings from specific areas to any specific cortical patterns (e.g. did lemniscal thalamic recordings have more similar STMs to SPMs vs non-lemniscal thalamic recordings?). Given that the correlation between cortical neurons and their neighbors is likely very different than the correlation between single thalamic neurons and their neighbors, it might be helpful to compute STMs for multiunit activity in cortex and in thalamus for a given electrode contact, and then subtract a scaled version of this STM from the single unit STM – that may put the cortical and thalamic analyses on more even footing (all the more true given data in Figure 3Diii). Similar analyses could be done for high-passed LFP or CSD recordings, allowing a better sense of the relationship between a single neuron's spike with network activity, vs. the correlated spiking of that neuron and many nearby neurons. This analysis may also reveal more spatially localized patterns (e.g. for barrel cortex recordings, it might reveal relatively more barrel-specific response profiles). Similarly, for cortical recordings, one could subtract STMs in superficial recordings from STMs with simultaneously recorded neurons in deep layers, thus removing the 'columnar' component of the correlation, and allowing evaluation of residual, layer-specific contributions.

2) Figure 8 is interesting in terms of differences between thalamic and cortical recordings, given the consistent net and prolonged suppression with thalamic STMs and net excitation with cortical STMs. While the authors mention Logothetis 2012 Nature, they don't mention that this Logothetis et al. paper (their Figure 3) shows hippocampal ripple-triggered excitation of most of cortex, and ripple-triggered suppression of most subcortical regions including thalamus. This could very well explain the effects seen in the current Figure 8, of net suppression in thalamic STMs (at most delays) and net activation for cortical STMs, as it provides evidence that cortical and thalamic neurons are likely anti-correlated during much of spontaneous activity (e.g. during ripple events). It would be worthwhile following this up – if most spontaneous cortical spiking is ripple-related and involved distributed, correlated cortical patterns, this could also partially explain why cortical STMs look more like SPMs than thalamic SPMs. This is a place where the true level and time course of activity would likely be different if STM analysis methods were used that didn't involve subtraction of GCaMP6 activity in the 3-s window prior to spike onset (see point 3c below).

3) Several analysis choices make the findings difficult to interpret: (a) there is no correction for vascular effects despite analyses spanning several seconds surrounding the spike: it is surprising that no vascular responses are seen in STMs in the GFP control, please discuss. (b) the authors suggest using deconvolution of signals as a future direction, but this would seem to be highly useful to use in the present analyses in separating GCaMp6 dynamics from neural dynamics, given the emphasis, in part, on temporal resolution (relative to fMRI). Both points (a) and (b) are effectively addressed in a study that is in some ways quite similar, by Wekselblatt et al., J Neurophys June 2016 -- a paper that is not cited by the authors. In that paper, Wekselblatt et al. nicely show the use of stimulus-evoked widefield GCaMP6 mapping across an intermediate sized cranial window in mice, after deconvolution and removal of vascular signals. (c) Another issue that is difficult to interpret is the normalization of GCaMP6 signal to a window of time prior to spike onset. Clearly this is throwing out a lot of useful information, and may distort much of the information that remains. An alternative approach might be to use a running median subtraction at each pixel, perhaps with removal of any slow drifts over long timescales, which would more effectively reveal what GCaMP6 activity is leading vs. lagging the spiking activity. (d) No discuss of the nature of the spiking activity is presented, yet burst firing (and thus, counting spikes with short ISI as similar to those with long ISI) may also distort the STMs. (e) The cortical delineation of areas is crude, and most secondary areas are not well delineated, making it difficult to assess whether any specific spatial resolution afforded by the technique actually yields insights at the level of specific higher cortical areas. (f): While it is interesting and useful to see that STM patterns become similar after averaging >128 or 256 trials (Figure 2), most of the information in the dataset is lost in this averaging process. Do subsets of spike events share a common spike-triggered pattern, albeit one that is not present on the majority of spike events, and which might be washed out by this averaging?

4) Description of the choice of epochs included in "spontaneous activity" is unclear. The authors do not seem to use body-tracking data that they collect and describe in Methods in order to segregate epochs of true spontaneous activity. The "spontaneous activity" may currently include sensory-evoked activity (i.e., internally generated spontaneous whisking), even though these behaviors were observed, as stated in "Methods: Multimodal recording in awake mice."

5) Figure 7 in interesting given the PCA patterns show differential connection with thalamic or cortical spiking, but no obvious effort was made to understand which cortical cells or thalamic cells fall into each pattern category (other than spike width analysis). What do spike autocorrelations look like? Sensory receptive fields? Layer in cortex? Location within specific thalamic nuclei? Without any such insights linking the findings to known entities, the value of this figure is somewhat low.

---

## [Author Response]

*Essential revisions:*

*Both reviewers as well as the Reviewing editor find this to be a carefully executed set of experiments and praise to quality of the imaging over an unprecedented nearly 1 cm by 1 cm field of view. The reviewers raise excellent issues and I would like the author to respond to all comments.*

We are pleased by the positive response of the reviewers and we have been able to address their collective and individual concerns as we describe below. As part of our revised manuscript, we now incorporate 11 new supplemental figures and 1 new video. We now hope the paper is ready for publication at *eLife*.

In summary, in response to the points highlighted as Essential Revisions: 1) we have addressed alternatives to averaging GCaMP signals and produced both variance and spike triggered co-variance maps, as well as devising a partitioning method to examine heterogeneity in single trials (see Figure 6—figure supplement 1–Figure 6—figure supplement 5). In the end, these approaches confirm the validity of an averaging approach; 2) Potential for changes of blood volume/oxygenation to alter fluorescence signals. We provide new data using a reference wavelength to confirm minimal effects of blood volume contamination of spike triggered average GCaMP signals (Figure 7—figure supplement 1); 3) We report that spike-triggered average maps are similar for bursting versus non-bursting neurons (Figure 6—figure supplement 1, Figure 6—figure supplement 2, Figure 6—figure supplement 3); and 4) We now show that periods of body movement have little impact on spike triggered average maps Figure 8—figure supplement 1).

Although not mentioned in the reviews we also take this opportunity to address verbal comments made by Matteo Carandini (at SFN during his lecture) about the Ai93 mouse and his observation of seizures. We are surprised by this observation as we have not seen activity consistent with the behavioral consequences of a seizure in our colony of these mice. We show long records of LFP activity indicating relatively normal up down states (Figure 9). We also show examples of spike waves that were generated after topical cortical application of a convulsant drug (picrotoxin). These results indicate that we can indeed detect seizures.

Author response image 1.Temporal and spectral signatures of spontaneous and epileptic events measured from the local field potential (LFP).(**A**) Top (black trace), 100s segment of LFP recording of spontaneous activity from superficial layer of cortex. Bottom, spectrogram (Morlet-wavelet scalograms) of the LFP trace. (**B**) Top (black trace), 170s segment of LFP recording after application of Picrotoxin (Sigma) on the top of cortical surface during the same experiment. (**i**) Top (black trace), 40s segment of LFP recording, epileptic events initiated with reversed phase and gradually increased amplitude (red arrows). Bottom, spectrogram of the LFP trace. (**ii**) Top (black trace), 40s segment of LFP recording showed repetitive high amplitude (>1 mV) epileptic discharge. Bottom, spectrogram showed high frequency power during the epileptic events but suppressed background activity.**DOI:**
http://dx.doi.org/10.7554/eLife.19976.029

*Regarding "essential" points, one reoccurring theme that will require additional analysis concerns the averaging. First, it would be very useful to see variance maps as well as average or spike triggered maps (STMs) as an extension of Figure 3 and Figure 4. The two may not be scaled versions of each other if the calcium responses are nonlinear. For that reason, the authors may further consider an analysis of the covariance of the data, as opposed to only the mean, as would reveal properties of "complex" versus "simple" visual cells. Perhaps this issue is at bay with the thalamic data. An example of how to do this was reviewed in Aljadef et al. (Neuron 2016), where the calcium images would replace the visual stimuli in the analysis.*

We thank the editor and reviewers for suggesting these extensions to the analysis. As suggested, we calculated spike triggered variance maps in an analogous way to the spike triggered maps and show an example of their temporal dynamics. While the spike triggered variance maps were of interest, spatially, they largely reveal familiar brain regions as defined using spike-triggered average (STA) mapping (when examined as a difference to baseline, see Figure 10). Considering the time course, there is a subtle, but global, shift to higher variance, but the variance is still very small (*△F/F0*from -0.2% to 0.2%) compared to the STA response.

Author response image 2.Spike-triggered variance map.**DOI:**
http://dx.doi.org/10.7554/eLife.19976.030

A. Spatiotemporal dynamic of spike triggered average map of a cortical neuron. The time window is from -3s to 3s. B.Spatiotemporal dynamic of spike triggered variance map of the same neuron. The amplitude of the variance is small (*△F/F_0_*(%) from -0.2 to 0.2).

The reviewer also makes a suggestion of examining co-variances making use of recently published methods (Aljadeff et al., 2016). We now include the spike-triggered covariance (STC) analysis as supplemental figures (Figure 11 and Figure 12). To do this we made use of the code provided in the supplemental information in Aljadeff et al. 2016 and used our calcium imaging in place of the visual stimuli, as suggested, following the analysis steps presented in this excellent resource. To make the computation feasible (in terms of computer time) it was necessary to down sample the imaging data to 32x32. In order to compare the resulting STA from the Aljadeff et al. 2016 with the spike triggered maps we previously calculated, we considered 6 time points. With these reduced dimensions it took 1-3 days to process a neuron depending on the length of the recording.

Author response image 3.Spike Triggered Covariance Analysis of BCS1 Neuron.(**A**) Each calcium image was recorded each Δt = 33.33 ms (frame rate = 30 Hz), and the spikes recorded in BCS1 neuron were binned within the same interval. In this example, we recorded 15250 frames and 4625 spikes simultaneously. (**B**) We constructed the covariance matrix of the spatiotemporal calcium images, followed by eigenvector analysis of the covariance matrix and plotted its spectrum (black). We compared this spectrum to that expected theoretically for the same-sized random matrix (red). (**C**) PCA of calcium images covariance matrix. The eigenvectors of the calcium images covariance matrix that correspond to the largest eigenvalues (mode 1 to 12) are seen to contain sequence of spatiotemporal patterns. (**D**) Spatiotemporal dynamic of STA map for cell #1 and cell#2. Calcium images were binned from 128*128 to 32*32 square pixel array with mask to select brain region. (**E**) Distribution of underlying calcium image matrix projected on Spike triggered average (STA) for cell #1 and cell #2. (**F**) The input/output nonlinearity of cell #1 and cell#2, input project on STA (S*sta).**DOI:**
http://dx.doi.org/10.7554/eLife.19976.031

Author response image 4.Spike Triggered Covariance Analysis of sub-cortical Neuron.(**A**) a1. STA of this sub-cortical neuron shows unique spatiotemporal pattern. a2. Distribution of underlying calcium image matrix projected on STA of this sub-cortical neuron. a3. The input/output nonlinearity of this neuron, input project on STA (S*sta). (**B**) b1. Spike triggered covariance mode 1 (STC1) of this neuron. b2. Distribution of underlying calcium image matrix projected on STC1. b3. The input/output nonlinearity of this neuron, input project on STC1 (S*stc_1_). (**C**) c1. Spike triggered covariance mode 2 (STC2) of this neuron. c2. Distribution of underlying calcium image matrix projected on STC2. c3. The input/output nonlinearity of this neuron, input project on STC2 (S*stc_2_). (**D**) The significance of each candidate STC feature, were determined by comparing the corresponding eigenvalue (red and black) to the null distribution (gray shaded area). We used 1,000 repetitions of the calculation for randomized spike trains, corresponding to a confidence level of 0.05. (**E**) Three dimensional plot of the nonlinearity in the space, spanned by the STA and other two orthogonalized STC feature, with surfaces for firing rates (FR) equal to 20% and 50% of the max.**DOI:**
http://dx.doi.org/10.7554/eLife.19976.032

Figure 11 (panel A) illustrates a montage of subsequent frames of calcium imaging with the recorded spikes as in Figure 3 of Aljadeff et al. 2016. Following Aljadeff et al. 2016 Figure 3, Figure 11 (panel B) shows the spectrum (eigenvalues) of the covariance matrix of the “stimulus” generated from our calcium imaging and Figure 11 (panel C) shows the spatial mode shapes. As expected, and in contrast to white noise, these have spatial structure and some familiar anatomical/functional regions are apparent. In Figure 11 (panels E & F) we show the underlying stimulus distributions and input/output nonlinearity curves for two cells (as in Aljadeff et al. 2016 Figure 4). The distributions can be expected to be Gaussian for a large n (central limit theorem, as pointed out in Aljadeff et al. 2016) and we would expect them to be more Gaussian-like if we were able to use our data at the recorded resolution (128x128). Figure 11 (panel D) shows the resulting STA and it compares well to our previously calculated spike triggered maps.

Continuing the analysis we next tested for significance of the STC features as in Aljadeff et al. 2016 Figure 5. We also used 1000 repetitions of random shifts of the spike train to generate null distributions. We first tested at a significance level of 0.01. At this significance level we were unable to find any significant STC feature for either cortical or subcortical neurons. The SNR of optical recordings may contribute to this so we re-tested at a more relaxed significance level of 0.05. At this level none of the n=4 cortical neurons tested had significant STC features. However, for 5 subcortical neurons of a total of n=5 tested we found significant STC features. Figure 12 illustrates this for a subcortical neuron similar to Aljadeff et al. 2016 Figure 5.

The analysis indicates that single cortical cells were most likely associated with a single dominant map, whereas single sub-cortical cells are not only associated with dominant STA map, but also more likely to be associated with multiple STC maps. While initial STC analysis indicated that BCS1 neurons were associated with more simple maps than subcortical neurons, GCaMP images may have had lower signal to noise ratio than some of the examples in Aljadeff et al. 2016 and we have considerably less spikes making it harder to draw firm conclusions from group analysis. Therefore, we show this analysis as a reviewer/editor-only example and have not included it as supplemental or regular figure.

*The reviewers raise three additional points that must be considered and preferably implemented. The first is the important issue of removing potential blood flow contributions, as was done by Chris Niell in a 2016 Journal of Neurophysiology manuscript. This should, ideally, be applied to the present data if at all possible.*

The reviewers mention that our data should be corrected for blood volume changes which can potentially alter conclusions regarding spike-triggered mapping.

All of the data in the original manuscript was collected with a single green epi-fluorescence wavelength and not dual wavelength imaging needed for a reflected light reference image. In the original paper we gave examples of GFP mice that would be subject to similar artifacts yet were unable to produced spike-triggered maps (Figure 2). As suggested by the reviewer these animals do show small slow transients that are typically less than 0.1% and importantly do not produce clear STA maps with known anatomical motifs.

To satisfy the reviewers concerns we have now done additional experiments using a multi-wavelength approach. We have now implemented a very similar correction to that performed by the Niell lab manuscript (Wekselblatt et al., 2016) and is also similar to work performed by the Hillman lab (Ma et al., 2016). In summary, the Niell lab implemented a strobed light presentation using alternating blue and green lights to capture GCaMP epi-fluorescence (blue illumination) and green reflectance (dim green illumination) in alternating frames. We have used a similar approach but monitor blue reflected light as our reference signal for fluorescence changes due to hemodynamics. We feel that there is some loss of time resolution using the strobed approach and it also depends very strongly on camera timing signals. There is also the issue of the flashing lights potentially stimulating/entraining the animal. Accordingly, we have developed a very similar strategy using a color RGB camera (Picam) (Murphy et al., 2016), which allows simultaneous acquisition of a short blue light reflected signal (447 nm LED and 438/24 nm filter near an isosbestic point for hemoglobin), and a green epi-fluorescence signal (GCaMP). Using the short blue reflected light signal, we now show that 438 nm reflected light is strongly correlated with 532 nm green reflected light in a control experiment (r = 0.93; Figure 7—figure supplement 1). This analysis indicates that short blue reflected light signals can be used as a surrogate for a green reflected signal. Accordingly, we now use the short blue reflected light signal in a division strategy very similar to previous approaches (Sirotin and Das, 2010; Ma et al., 2016; Wekselblatt et al., 2016). The division was performed after both frame-by-frame signals were converted a ΔF/F_0_ and employed a green/blue weight of 1. We have tried weightings than >1.0 for the short blue signal, but found that some aspects of the kinetics were over-corrected. This new analysis dampens non-specific fluctuations associated with blood volume changes, which are aggravated in the awake state. We now show using validation with green light reflection, as well as GFP mice and that our strategy results in a reduction in baseline noise.

However, after implementing this corrective strategy, we find little change in spike-triggered map activity consistent with control investigations of functional connectivity or task-related connectivity in GFP animals in experiments done previously by our lab (Vanni and Murphy, 2014; Murphy et al., 2016; Silasi et al., 2016). Furthermore, notable features, such as some areas showing apparent cortical inhibition (reductions in calcium activity) were also preserved in the new corrected analysis. We now include this new data as a supplemental figure (Figure 7—figure supplement 1) and outline the procedure in the Materials and methods and Results section. Given that the correction does not have a large impact and failed to change the appearance of the maps we have kept the original figures as is (they cannot be corrected given they reflect a single wavelength) and now point readers to the new confirmatory experiments and previous papers using the GFP-mouse controls (Vanni and Murphy, 2014; Murphy et al., 2016; Silasi et al., 2016).

See amended text from Materials and methods:

“In some experiments (as indicated), we employed a multi-wavelength strategy to correct for potential green epifluorescence signals that were associated with non-calcium dependent events. […]While the shorter blue wavelength light will scatter more than a green strobed reflected light signal used by others (Ma et al., 2016; Wekselblatt et al., 2016), our analysis of green reflected light and short blue reflected light, indicating that two were highly correlated, suggests that the major artifacts we observe are associated with large blood volume changes in superficial cortical layers.”

See amended text from Results section:

“Other potential sources of error include apparent changes in GCaMP6 signal due to alterations in blood volume or oxygenation (Ma et al., 2016; Wekselblatt et al., 2016). […]Furthermore thalamic STA maps and STA dynamic plots still indicated cases where thalamic spiking was associated with cortical inhibition. For these reasons we have attempted to correct other STA maps using this multi-wavelength procedure.”

*The second concerns the nature of the STM calculated from temporally isolated spikes versus spikes that are part of a burst. Given the known depression of thalamo-cortical synapses, this could likely contribute the variability of the STM data using thalamic spikes. To address this issue, the thalamic bursts can be isolated from the spike records and, as one possibility, the STM can be computed with respect to only the first spike.*

It is conceivable that averaging over many spikes could mask spatial heterogeneity observed in sub-groups of spikes. Therefore, we performed additional analyses where we classified spike triggered STMs and STMTDs into sub-groups based on spiking modes (bursting vs. tonic) or motif similarity in a high-dimensional space – with the goal of determining whether during different types of active cortical dynamics our single spike-sorted neurons contribute different types of STMs or temporal dynamics (Figure 6—figure supplement 1–Figure 6—figure supplement 5).

We searched for STM sub-groups by dividing all spikes from a single cell into bursting versus tonic modes. We implemented methodology previously described for defining the main spiking modes of a single cell (Sherman and Guillery, 2006; Figure 6.5, pg 236). Briefly, the method requires computing the distribution of each spike’s inter-spike-interval (ISI) between the previous (x-axis) and following (y-axis) spike (Figure 6—figure supplement 1). The resulting pre- and post-spike 2D distribution is then plotted using logarithmic scales and naturally arising clusters are grouped or clustered. Spike groups occurring in approximately each quadrant of the plot indicate different spiking modes: first spikes in a burst (bottom right), spiking occurring during a burst (bottom left), last spikes in a burst (top left), and tonic spikes (top right). The vast majority of cortical cells we recorded in barrel cortex did not exhibit multiple classes of spiking modes, however we provide 2 examples where some natural clustering is present (Figure 6—figure supplement 1(Ai) and (Bi)). The spike sub-groups representing different spiking modes yielded very similar STMs and motifs (Figure 6—figure supplement 1(Aii) and (Bii)) and similar STMTDs in the area of highest activation, i.e. left barrel cortex (Figure 6—figure supplement 1 (Aiii) and (Biii)). We also provide examples of two identified thalamic cells which showed more apparent spiking modes (Figure 6—figure supplement 1(C) and (D)). The thalamic cells we identified showed better bursting versus tonic clusters (Figure 6—figure supplement 1 (Ci) and (Di)) but much like cortical cells their STM and STMTDs were largely stable across the clustered spiking modes. The 3 cortical and 4 thalamic cells provided in Figure 6 did not exhibit clear spiking modes. For those cells we implemented a simpler methodology and compared all-spike STMs against STMs generated by spikes preceded by at least 500ms of silence as indicative of bursting activity following a period of quiescence (Figure 6—figure supplement 2 and Figure 6—figure supplement 3). Using both of these methodologies to divide spiking times into different modes we found that both STMs and STMTDs across spiking modes are similar to the all-spike average STMs in both cortical and thalamic cells examined.

We additionally sought to determine whether single-spike motifs were similar to each other and formed natural clusters in a high-dimensional representation space. There were no naturally arising clusters suggesting that single spikes occur during many phases of ongoing, i.e. spontaneous, cortical activity. Accordingly, we manually partitioned the motifs into several (4 to 12) groups by similarity and after removing the spontaneous component we were able to recover the all-spike average STMs (Figure 6—figure supplement 4 and Figure 6—figure supplement 5; see Methods). This single-spike motif sub-grouping suggests that: (i) cells fire during many different ongoing cortical states; and (ii) that despite the high degree of variability single cell spikes appear to correlate with (or possible contribute to) similar overall STM patterns. Given that single spike motifs are very diverse (Figure 6—figure supplement 4(A)) our approach was to project single-spike STMs into a high-dimensional space, compute their distributions and search for clusters or manually partition the resulting distributions. There were no natural groups in the resulting high-dimensional distribution so we partition the data into 4 (or more) sub-groups (Figure 6—figure supplement 4(B), see also Figure 6—figure supplement 5). The resulting 4 “sub-networks” represent (on average) qualitatively different types of ongoing cortical activity during which our cell fires action potentials. While these partitions were manually created, we were interested in determining whether they could generate STMs that resembled all-spike average STMs and whether this was independent of the number of partitions. The 4 partitions chosen generated somewhat different STMs, only one of which strongly resembled our all-spike average STM (Figure 6—figure supplement 4(D); all spike average is bottom STM). The inter-spike-interval (ISI) distributions during activation of these sub-networks is also similar across all partitions further suggesting that this partitioning method did not identify highly bursting or tonic periods of activity (Figure 6—figure supplement 4(C)). Because we had enforced a manual STM-space partition, our resulting sub-network STM averages contained an offset that essentially represented ongoing “spontaneous” cortical activity particular to each STM-space partition. Accordingly, we sought to remove this spontaneous component by carrying out the same partitioning method – but this time on spontaneous activity (i.e. not using single spike triggers). The distribution of spontaneous STMs (i.e. ongoing STMs not related to our cell spiking) also have substantial variability (Figure 6—figure supplement 4(E)). We next projected all spontaneous motifs into an STM-space but partitioned this space based on the sub-network centres derived previously (Figure 6—figure supplement 4(B) and (D)). This guaranteed that the 4 STMs derived from spontaneous activity would be the closest to our spike-triggered STMs. The spontaneous activity motifs showed a very strong ISI distributions peak at ~33ms (i.e. single-frame time) (Figure 6—figure supplement 4(G)) indicating that STMs that were neighbouring in time (i.e. separated by single frames) were substantially more likely to be in the same region of STM-space and to be grouped together. This is expected as transitions meso-scale cortical dynamics usually last several frames at our 30Hz sampling rate. The sum of the 4 spontaneous STMs yield an STM with an approximately 0% ΔF/F_0_ value – which is expected when averaging all spontaneous motifs (or frames) during a recording (Figure 6—figure supplement 4(H) bottom STM). The last step is to remove the spontaneously activity STMs (Figure 6—figure supplement 4(H)) from the single cell spike-triggered sub-networks (Figure 6—figure supplement 4(D)). The results reveal that the 4 partition STMs reduce to the all spike average STMs. This analysis supports the averaging method of computing STMs: despite firing during different types of cortical dynamics, single cells contribute to – and participate in – ongoing dynamics in a stereotyped way that is represented by the all-spike STM average.

We further applied this partitioning method on additional cells: two cortical and two thalamic cells with similar results (Figure 6—figure supplement 5). We also tested a 12 partition approach (Figure 6—figure supplement 5(C)) and showed that when subtracting spontaneously activity STMs (Figure 6—figure supplement 5(C)(ii)) from the spike triggered STM (Figure 6—figure supplement 5(C)(i)) we obtain STMs that are very similar to the all-spike averages (Figure 6—figure supplement 5(C)(iii)). This generally confirms that average based STMs are representative of a single cell’s contribution to ongoing cortical activity.

See amended text from Materials and methods:

“Cell Spiking Mode Determination. We implemented methodology previously described for defining main spiking modes (Sherman et al., 2006) (Figure 6.5, pg 236). […]We provide additional examples of this STM partitioning using other cortical and thalamic cells and also an example where we partitioned the STM distribution into 12 sub-networks but still recovered the all-spike average STM from each of the partitioned networks (Figure 6—figure supplement 5).”

See amended text from Results section:

“Our analysis indicates that averaging GCaMP cortical motifs from all spikes of a single cell produces converging STMs and STMTDs. […]These tests confirm that the averaging method reveals how single cells participate in largely stereotyped networks despite the variability of ongoing cortical activity.”

*Lastly, the "spontaneous activity" may currently include sensory-evoked activity from body motion. As the authors recorded body motion, it would be germane to check if "spontaneous activity" in fact results from self motion, as this changes the interpretation of "spontaneous activity' from internally generated to reafferent driven.*

Mice when habituated to our head-fixed recording apparatus do not exhibit extensive active body movements. However, the reviewer’s point is well-taken and we explored the potential contribution of active movements, however minute, to our recordings of spontaneous data. In new analysis, we now use simultaneously acquired video imaging (webcam) data to segment quiet periods of animal activity for spike-triggered averaging. Despite this procedure, we found little difference in spike-triggered mapping when movement periods were removed from awake mouse spontaneous activity data (Figure 8—figure supplement 1 and Video 5).

See amended text from Materials and methods:

“While relatively few large body movements were observed during awake imaging sessions, their impact on mapping was evaluated by generating STA from period of quietness. […] STA was then generated only from spikes for periods of quietness and compared with STA of all spikes.”

Amended section from Results section:

“Our goal is to assess cortical functional connectivity based on coincidence between individual neuron spiking and ongoing spontaneous activity. […] Therefore, brain imaging activity obtained in awake states is mostly indicative of a quiet awake state and is not primarily movement-related activity.”

Reviewer #1:

*[…] Major issues:*

*1) Results section. Spike-trigger averaging of optical signals was originally reported by Grinvald lab and authors are simply adopting this approach to the large-scale GCaMP imaging. Authors need to introduce the original work in the Introduction and/or Results section.*

Although we have referenced these papers, including within the Introduction, we agree that we could do more to highlight their seminal contributions that include the use of early application of spike-triggered averaging wide field imaging methods and conclusions. We have now revised our Introduction section to better acknowledge this pioneering work.

The amended Introduction now states:

This work extends pioneering studies investigating the relationship between single neuron spiking and local neuronal population activity assessed by voltage-sensitive dye imaging. […] Furthermore, we employ multisite, long shank, silicon probe recordings of single neuron activity that facilitates the assessment of long-distance activity relationships between remote subcortical single neuron activity and mesoscale cortical population activity.

We also do note differences between our GCaMP wide-field mouse approach and the previous work in visual cortex. Notable differences include the neuronal specificity of the GCaMP imaging signals, as well relatively wider field of view and additional information concerning sub-cortical spiking neurons.

*2) Figure 1. Authors are using single-spike events to calculate the STM. However, in GCaMP imaging, single spikes can produce only marginal response changes (*Δ*F/F). In addition, a single spike in a single neuron is unlikely to effectively activate all the post-synaptic neurons. Thus, it is possible that multiple-spike events (or bursts) can produce a more reliable STM. The authors could test this possibility by reanalyzing data and the results could be shown as in Figure 2.*

The reviewer raises an important point regarding the potential nuanced relationship between the pattern of cell firing and the efficacy in producing a reliable STM. The reviewer suggests that in the case of GCaMP imaging, individual spikes may not go readily detected. We reference papers which show that GCaMP6 can detect individual spikes in vivo (Chen et al., 2013). However, we feel we must clarify that our intention is not to imply a causal relationship between the unitary contributions of a single cell’s spiking activity to mesoscale cortical calcium activity or that our method requires the faithful detection of individual spikes. Instead, we interpret the single cell spiking activity as being correlated with the activation of these large cortical networks and likely acting in synchrony with other cells. However, the issue of averaged cell firing pattern diversity and its impact on STMs is important and explored in greater detail in our response to the Essential Revisions Comment #3.

*3) Figure 2. Authors demonstrate that a conserved network motif emerges by averaging the spike-triggered maps. However, this is based on an assumption that a single spike is always linked to a single type of network motif. Is it possible that multiple motifs are linked to a single neuron and authors' approach is averaging out multiple types of motifs? Authors could, for example, try classification of the spike-triggered maps.*

The reviewer raises an excellent point and it is not our intention to imply that a single spike is always linked to a single type of network motif but rather that the average response yields a pattern consistent with these consensus maps. However, one of our principle conclusions is that numerous, diverse sub-networks of cortical activity patterns exist and can be linked with the activity of a single cell as illustrated in Figure 6—figure supplement 1 to 5). The discussion of averaging and heterogeneity and our new analyses are described in detail in our response to the Essential Revisions Comment #3.

*4) Figure 3. Given the diversity of cell types (and projection patterns) within a defined brain region, it is possible that STM is diverse even within a defined cortical area. Did authors find such cases? (In fact, STMTD is diverse in Figure 7 and 8.) It is also possible that neurons in different cortical layers show distinct STMs even within the same area. Throughout this manuscript, box plots or beeswarm plots may be more appropriate if they show non-normal distributions with rare examples.*

As mentioned, given the strong agreement between seed pixel maps and single neuron spike-triggered maps in cortex, we do not expect a lot of averaged-STM diversity. In the original submitted manuscript we provided “contour” plot analysis and quantification of individual STM maps across laminar depth (Figure 3). We found that they were largely super-imposed and somewhat invariant across cortical depth. In the revised manuscript, we better highlight these findings with new analyses where we illustrate the STMs generated by single unit activity, multi-unit activity, and LFP across laminar depths from cortical and subcortical recordings (Figure 3—figure supplement 1 and Figure 3—figure supplement 2).

Another consideration is that we insert the electrode using an angular tilt (45°) to achieve an approximately vertical penetration to the curved cortical surface such that the electrode is roughly recorded in the same functional cortical column. In the same functional column, the single neuron related functional maps may be more likely similar, even there are diverse cell types. We have also replaced the bar graph in Figure 4 with a box plot.

*Reviewer #3:*

*[…] 1) The authors show that seed pixel correlation maps (SPMs) are better matched to cortical single-unit spike-triggered maps (STMs) than to thalamic STMs, and that thalamic STMs are generally more diverse than cortical (specifically barrel cortex) STMs. However, as suggested by the authors in the Discussion, this is likely due, at least in part, to greater diversity across thalamic recordings from neurons in a larger number of smaller thalamic nuclei (vs. more consistently from S1 barrel region of cortex, albeit across layers) vs. recordings in a single cortical area with a large magnification factor. Further, cortical neurons within S1 are highly correlated, likely more so that nearby neurons in thalamus, further ensuring strong correlation between the cortical spiking activity and the SPM maps (since the seed cortical pixel is likely highly correlated with the cortical single-unit recording), more so that would be likely be possible with most thalamic units). The other suggested explanations for these findings made by the authors in the Discussion are far more speculative, and many would be difficult to test. Further, little effort is made to relate specific thalamic recordings from specific areas to any specific cortical patterns (e.g. did lemniscal thalamic recordings have more similar STMs to SPMs vs non-lemniscal thalamic recordings?). Given that the correlation between cortical neurons and their neighbors is likely very different than the correlation between single thalamic neurons and their neighbors, it might be helpful to compute STMs for multiunit activity in cortex and in thalamus for a given electrode contact, and then subtract a scaled version of this STM from the single unit STM – that may put the cortical and thalamic analyses on more even footing (all the more true given data in Figure 3Diii). Similar analyses could be done for high-passed LFP or CSD recordings, allowing a better sense of the relationship between a single neuron's spike with network activity, vs. the correlated spiking of that neuron and many nearby neurons. This analysis may also reveal more spatially localized patterns (e.g. for barrel cortex recordings, it might reveal relatively more barrel-specific response profiles). Similarly, for cortical recordings, one could subtract STMs in superficial recordings from STMs with simultaneously recorded neurons in deep layers, thus removing the 'columnar' component of the correlation, and allowing evaluation of residual, layer-specific contributions.*

Reviewer 3 is generally positive about the work and appreciates the linkage between wide-scale cortical activity and studies which have examined event-related activation using fMRI imaging.

The reviewer makes a number of good points, mostly about better examining the relationship between individual neurons and the ensemble brain activity.

1) The reviewer wonders about our interpretation of findings which compare diversity in cortical spike-triggered GCaMP maps for spike trains produced by cortical versus sub-cortical neuron groups. They acknowledge our discussion points, including diversity being attributed to the physical layout of the thalamus versus cortex and the ability to record from potentially more diverse neurons in the thalamus given the relatively smaller structure over which things are mapped. Accordingly, they suggest experiments to place subcortical thalamic neurons on more even footing with cortical neurons. These include making a spike-triggered single cortical neuron map in which multi-unit activity or LFP-triggered activity has been subtracted away. This would potentially allow the contributions of individual cortical neurons to be more easily seen.

This is a great suggestion and we have already done preliminary studies using single-channel MUA spiking (Figure 3—figure supplement 1 and Figure 3—figure supplement 2). We also computed band-passed LFP triggered STMs for Delta (0.1-4Hz), Theta (4-8Hz) and Gamma (25-100Hz). LFP triggered STMs were computed using 60 second periods by multiplying the average LFP amplitude in each imaging time frame. Thus, imaging frames where the average LFP amplitude was large and positive contributed substantially to the STM, whereas frames where the average LFP values were closer to zero did not (negative LFP values were clipped). The findings for both cortical and subcortical recordings are that MUA triggered STMs are similar to single-unit STMs in cortex while LFP triggered STMs – representing mostly synaptic and some spiking activity, were substantially different from single cell and MUA triggered STMs and across different LFP frequency bands. However, it is important to note that LFP amplitude (or power-) -triggered maps have very low ΔF/F values, for example most have peak ΔF/F values < 0.05% (whereas single-unit or MUA peaks are between 1%-5%). Nonetheless, cortical LFP triggered maps do reveal anatomically discrete maps (Figure 3—figure supplement 1). However, while there appear to be some slight differences across layers, the differences do not appear to be substantial or systematic. It is likely that as LFP contains mostly synaptic and some spiking activity from within a region with radius of at least 250μm (Buzsaki et al., 2012) the STMs may represent average activity from larger neighbouring regions thus blurring the overall effect of local LFP activity.

In response to the reviewer’s suggestions that we subtract “a scaled version” of MUA based STM from single-unit STM, unfortunately the results appear largely as very noisy maps (i.e. maps with <1% ΔF/F values that have no ROI specificity). This is due to the similarities between single-unit and MUA STMs before subtraction. However, we agree overall that this general direction could be pursued further in future work to determine whether the limit of the GCaMP temporal and spatial resolution allows for more nuanced single-unit STMs to be obtained by removing common baselines.

See amended text from Results section:

“We have also assessed other means of generating event triggered maps including MUA and LFP frequency bands. […] In contrast, in the case of GCaMP6 higher frequency components are closer to hemodynamic and other noise sources making analysis more challenging.”

*2) Figure 8 is interesting in terms of differences between thalamic and cortical recordings, given the consistent net and prolonged suppression with thalamic STMs and net excitation with cortical STMs. While the authors mention Logothetis 2012 Nature, they don't mention that this Logothetis et al. paper (their Figure 3) shows hippocampal ripple-triggered excitation of most of cortex, and ripple-triggered suppression of most subcortical regions including thalamus. This could very well explain the effects seen in the current Figure 8, of net suppression in thalamic STMs (at most delays) and net activation for cortical STMs, as it provides evidence that cortical and thalamic neurons are likely anticorrelated during much of spontaneous activity (e.g. during ripple events). It would be worthwhile following this up – if most spontaneous cortical spiking is ripple-related and involved distributed, correlated cortical patterns, this could also partially explain why cortical STMs look more like SPMs than thalamic SPMs. This is a place where the true level and timecourse of activity would likely be different if STM analysis methods were used that didn't involve subtraction of GCaMP6 activity in the 3-s window prior to spike onset (see point 3c below).*

We thank the reviewer for highlighting these findings from Logothetis et al. As we did not record simultaneously from the hippocampus we cannot definitely comment on the relationship between thalamic spiking activity and the observation of net cortical inhibition with respect to hippocampal ripples in our experiments. However, the data presented in Logothetis et al. are remarkable and consistent with this interpretation and we now more clearly discuss our observations in this interesting context in our revised Discussion.

See amended text in Discussion section:

“The approach is probably most analogous to event-triggered MRI imaging from the standpoint of larger spatial scale (Logothetis et al., 2012). […] However, as we did not record simultaneously from hippocampus we cannot definitively comment on whether these observations necessarily correspond to hippocampal ripple events.”

With respect to the specific comment of “the true level and time course” of activity, we have addressed this in greater detail in regard to the reviewer suggestion of implementing a running median subtraction in comment (3C).

*3) Several analysis choices make the findings difficult to interpret: (a) there is no correction for vascular effects despite analyses spanning several seconds surrounding the spike: it is surprising that no vascular responses are seen in STMs in the GFP control, please discuss. (b) the authors suggest using deconvolution of signals as a future direction, but this would seem to be highly useful to use in the present analyses in separating GCaMp6 dynamics from neural dynamics, given the emphasis, in part, on temporal resolution (relative to fMRI). Both points (a) and (b) are effectively addressed in a study that is in some ways quite similar, by Wekselblatt et al., J Neurophys June 2016 -- a paper that is not cited by the authors. In that paper, Wekselblatt et al. nicely show the use of stimulus-evoked widefield GCaMP6 mapping across an intermediate sized cranial window in mice, after deconvolution and removal of vascular signals.*

The reviewer notes that there is no correction for potential vascular effects despite the signals spanning several seconds around the spike and it is surprising that we do not see vascular responses in the spike-triggered maps from GFP controls. We have more thoroughly addressed this point in our response to the Essential Revisions (comment 2) and now report new data using a normalization procedure in a new supplemental figure (Figure 7—figure supplement 1). In our characterization of this strategy, we have also done additional GFP mice. In no case do we see a clear reduction in baseline signal. Perhaps this is because spike-triggered maps are relatively discreet events and are not associated with global changes in activity which could potentially recruit hemodynamic responses. Another issue the reviewer brings up is the normalization of GCaMP signals to an average time window. We should note that in the spike-triggered average movies and images of temporal dynamics that we do indeed show imaged which reflect relative changes in ΔF over F during the preceding time. We have also attempted to implement the median subtraction approach.

The reviewer suggests a temporal deconvolution for signals to improve time resolution. We have now performed such an analysis and report these results (Figure 1—figure supplement 2). While sharpening the kinetics for raw data the procedure does not significantly change maps and it was not applied to all data.

*(c) Another issue that is difficult to interpret is the normalization of GCaMP6 signal to a window of time prior to spike onset. Clearly this is throwing out a lot of useful information, and may distort much of the information that remains. An alternative approach might be to use a running median subtraction at each pixel, perhaps with removal of any slow drifts over long timescales, which would more effectively reveal what GCaMP6 activity is leading vs. lagging the spiking activity.*

The reviewer suggests different methods for averaging calcium activity frames, in particular a “running median” method. We previously employed two different methods including a “global signal regression” method in which the baseline fluorescence is the mean of the entire ~5 minute recording stack. We also employed a low-band pass filtering method where the entire image stack was filtered at 0.1Hz-6Hz. The STMs provided by these methods appeared to be largely similar to the sliding window methodology likely because averaging over 3sec, 5 minutes or low-band pass filtering yield similar components in the calcium signal.

Slow drifts were very occasionally observed during recordings. However, in the initial manuscript, STA generated from random spike times, not carrying neuronal activity but still affected by slow drifts in the same manner, were used for correction (by subtraction, see Figure 13). The use of running median gives the same results on slow drift but in contrast to random spike STA, a median window has to be determine and if inappropriately set, could remove important components such as depression. For this reason this normalization strategy, while otherwise elegant, was avoided.

Author response image 5.Running median and random spike normalization.(**A**) STA from real spikes (1^st^ line), random spikes (2^nd^ line) and real spikes after running median filtering (3^rd^ line). 4^th^ and 5^th^ lines are subtractions of STA by random spikes and running median respectively. (**B**) STA dynamic within BCS1 for real spikes (black curve), random spikes (thin red) and the subtraction (Thick red). (**C**) Same as B but for running median. (**D**,**E**) While drift was not often observed it was artificially introduced by reducing the DC of spontaneous activity by 50% in 9min. Both subtractions (random spikes and running median) were able to remove the slow drift contribution. However, post-spike depression (black arrow) at ∼1s was only preserved by using random spikes correction.**DOI:**
http://dx.doi.org/10.7554/eLife.19976.033

*(d) No discuss of the nature of the spiking activity is presented, yet burst firing (and thus, counting spikes with short ISI as similar to those with long ISI) may also distort the STMs.*

We have now addressed the issue of the potential relationship of burst firing and STMs in our response to the Essential Revisions Comment #3.

*(e) The cortical delineation of areas is crude, and most secondary areas are not well delineated, making it difficult to assess whether any specific spatial resolution afforded by the technique actually yields insights at the level of specific higher cortical areas.*

The reviewer would like to see better delineation of secondary cortical areas. Unfortunately, using a bilateral imaging window implant, we do not have optical access to most secondary somatosensory areas. We have replaced the figure panel in Figure 1 with an enlarged and more clearly defined cortical reference atlas and updated table of regions-of-interest and abbreviations used to better elucidate our imaging perspective.

Regarding spatial resolution, we have revised the Methods section to include acknowledgement of the spatial resolution of our imaging and references to our previous work from which these measurements are described in detail.

See the amended text in Materials and methods:

“In order to reduce file size and minimize the power of excitation light used, we typically bin camera pixels (8 × 8) thus producing a resolution of 68 µm/pixel. These imaging parameters have been used previously for voltage sensitive dye imaging (Mohajerani et al., 2013) as well as anesthetized GCaMP3 imaging of spontaneous activity in mouse cortex (Vanni and Murphy, 2014) and awake GCaMP6 imaging in mouse cortex with chronic window (Silasi et al., 2016).”

*(f): While it is interesting and useful to see that STM patterns become similar after averaging >128 or 256 trials (Figure 2), most of the information in the dataset is lost in this averaging process. Do subsets of spike events share a common spike-triggered pattern, albeit one that is not present on the majority of spike events, and which might be washed out by this averaging?*

We have now addressed the issue of heterogeneity of STMs and the use of averaging in our response to the Essential Revisions Comment #3.

*4) Description of the choice of epochs included in "spontaneous activity" is unclear. The authors do not seem to use body-tracking data that they collect and describe in Methods in order to segregate epochs of true spontaneous activity. The "spontaneous activity" may currently include sensory-evoked activity (i.e., internally generated spontaneous whisking), even though these behaviors were observed, as stated in "Methods: Multimodal recording in awake mice."*

We have now addressed the issue of the potential contribution of active behaviours in our response to the Essential Revisions Comment #4.

*5) Figure 7 in interesting given the PCA patterns show differential connection with thalamic or cortical spiking, but no obvious effort was made to understand which cortical cells or thalamic cells fall into each pattern category (other than spike width analysis). What do spike autocorrelations look like? Sensory receptive fields? Layer in cortex? Location within specific thalamic nuclei? Without any such insights linking the findings to known entities, the value of this figure is somewhat low.*

Our results presented in Figure 7 indicate that single cortical and thalamic cells participate in ongoing contralateral barrel cortex activating/depressing dynamics that fall into 2 or 3 discrete groups, respectively. The reviewer’s question of whether this grouping correlates with other single cell properties or whether it may be a novel intrinsic single cell property is very interesting but we were not able to identify correlations given our available datasets. Our cell-type, i.e. inhibitory versus excitatory, classification did not show correlation between PCA clusters and cell type. Additionally, we did not carry out stimulus paradigms to allow us to compute receptive fields for our barrel cortex cells. However, we did not find a correlation between cortical layers and PCA clusters nor between VPM and VPL nuclei and specific PCA clusters. Lastly, we did not pursue spike autocorrelation analysis as it would require classification or grouping of autocorrelation results which would go beyond the scope of our work. We anticipate that further work with substantially larger numbers of single cells may better classify activation/depression dynamics and allow for the use of multiple ROIs (e.g. barrel, motor, retrosplenial). We agree that the alternative, i.e. that single cells may have intrinsic properties that bias them to participating in networks acting on substantially different temporal scales, is certainly a topic worth investigating further.

See amended text in Results section:

“This classification showed that ~80% cortical neurons were associated with a pure cortical excitation profile (pattern #1) and 20% triggered inhibition (pattern #2). […] We suggest future work with larger datasets and multiple cortical/subcortical areas could address the question of whether STMTD classification are a novel intrinsic single-cell property.”